# Label Leakage and Protection in Two-party Split Learning

**Oscar Li**[1][†][*]**, Jiankai Sun**[2][§]**, Xin Yang**[2]**, Weihao Gao**[2]**,**
**Hongyi Zhang**[2]**, Junyuan Xie**[2]**, Virginia Smith**[1]**, Chong Wang**[2]

[1]Carnegie Mellon University
[2]ByteDance Inc.

## Abstract

Two-party split learning is a popular technique for learning a model across feature-partitioned data. In this work, we explore whether it is possible for one party to steal the private label information from the other party during split training, and whether there are methods that can protect against such attacks. Specifically, we first formulate a realistic threat model and propose a privacy loss metric to quantify label leakage in split learning. We then show that there exist two simple yet effective methods within the threat model that can allow one party to accurately recover private ground-truth labels owned by the other party. To combat these attacks, we propose several random perturbation techniques, including `Marvell`, an approach that strategically finds the structure of the noise perturbation by minimizing the amount of label leakage (measured through our quantification metric) of a worst-case adversary. We empirically[1] demonstrate the effectiveness of our protection techniques against the identified attacks, and show that `Marvell` in particular has improved privacy-utility tradeoffs relative to baseline approaches.

## 1 Introduction

With increasing concerns over data privacy in machine learning, *federated learning* (FL) ([McMahan et al., 2017](#)) has become a promising direction of study. Based on how sensitive data are distributed among parties, FL can be classified into different categories, notable among which are *horizontal FL* and *vertical FL* ([Yang et al., 2019](#)). In contrast to horizontal FL where the data are partitioned by examples, vertical FL considers data partitioned by features (including labels). As a canonical example of vertical FL, consider an online media platform $A$ which displays advertisements from company $B$ to its users, and charges $B$ for each *conversion* (e.g., a user clicking the ad and buying the product). In this case, both parties have different features for each user: $A$ has features on the user's media viewing records, while $B$ has the user's conversion labels. $B$'s labels are not available to $A$ because each user's purchase behaviors happen entirely on $B$'s website/app.

If both parties want to jointly learn a model to predict conversion without data sharing, *split learning* ([Gupta & Raskar, 2018](#); [Vepakomma et al., 2018](#)) can be used to split the execution of a deep network between the parties on a layer-wise basis. In vanilla split learning, before training begins, both parties use Private Set Intersection (PSI) protocols ([Kolesnikov et al., 2016](#); [Pinkas et al., 2018](#)) to find the intersection of their data records and achieve an example ID alignment. This alignment paves the way for the split training phase. During training (Figure [1](#)), the party without labels (*non-label party*) sends the intermediate layer (*cut layer*) outputs rather than the raw data to the party with labels (*label party*), and the label party completes the rest of the forward computation to obtain the training loss. To compute the gradients with respect to model parameters, the label party initiates backpropagation from its training loss and computes its own parameters' gradients. To allow the non-label party to also compute gradients of its parameters, the label party also computes the gradients with respect to the cut layer outputs and communicates this information back to the non-label party. As a result of the ID alignment, despite not knowing the label party's raw label data, the non-label party can identify the gradient value returned by the label party for each example.

At first glance, the process of split learning appears privacy-preserving because only the intermediate computations of the cut layer—rather than raw features or labels—are communicated between the two parties. However, such "gradient sharing" schemes have been shown to be vulnerable to privacy leakage in horizontal FL settings (e.g., [Zhu et al., 2019](#)). In **vertical FL** (and specifically split learning), it remains unclear whether the raw data can similarly be leaked during communi-

---

[*]work done during internship at ByteDance Inc. [†]oscarli@cmu.edu [§]jiankai.sun@bytedance.com

[1]Code available at https://github.com/OscarcarLi/label-protection

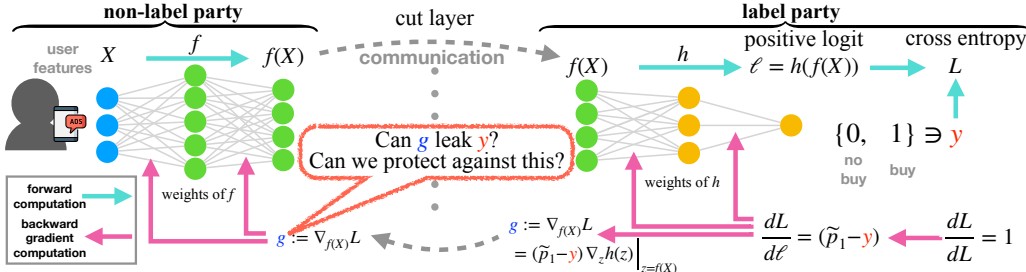

Figure 1: Communication diagram of two-party split training for an example of online advertising. We study whether it is possible for the communicated gradient $g$ to leak private label information.

cation. In particular, as the raw labels often contain highly sensitive information (e.g., what a user has purchased (in online advertising) or whether a user has a disease or not (in disease prediction) Vepakomma et al. (2018)), developing a rigorous understanding of the threat of label leakage and its protection is particularly important. Towards this goal, we make the following contributions:

1. We formalize a threat model for label leakage in two-party split learning in the context of binary classification (Section 3.1), and propose specific privacy quantification metrics to measure the severity of such threats (Section 3.2).

2. We identify two simple and realistic methods within this threat model which can accurately recover the label party's private label information (Section 3.3).

3. We propose several random perturbation techniques to limit the label-stealing ability of the non-label party (Section 4). Among them, our principled approach Marvell directly searches for the optimal random perturbation noise structure to minimize label leakage (as measured via our quantification metric) against a worst-case adversarial non-label party.

4. We experimentally demonstrate the effectiveness of our protection techniques and MARVELL's improved privacy-utility tradeoffs compared to other protection baselines (Section 5).

## 2 RELATED WORK

**Privacy leakage in split learning.** Although raw data is not shared in federated learning, sensitive information may still be leaked when gradients and/or model parameters are communicated between parties. In horizontal FL, Zhu et al. (2019) showed that an honest-but-curious server can uncover the raw features and labels of a device by knowing the model architecture, parameters, and communicated gradient of the loss on the device's data. Based on their techniques, Zhao et al. (2020) showed that the ground truth label of an example can be extracted by exploiting the directions of the gradients of the weights connected to the logits of different classes. Here we study a different setting—two-party split learning (in vertical FL) (Yang et al., 2019), where no party has access to the model architecture or model parameters of the other party. In this setting, Vepakomma et al. (2019) studied how the forward communication of feature representations can leak the non-label party's raw data to the label party. We instead study whether *label* information may be leaked from the label party to the non-label party during the backward communication. Despite the importance of maintaining the privacy of these labels, we are unaware of prior work that has studied this problem.

**Privacy protection and quantification.** Techniques to protect communication privacy in FL generally fall into three categories: **1)** cryptographic methods such as secure multi-party computation (e.g., Bonawitz et al., 2017); **2)** system-based methods including trusted execution environments (Subramanyan et al., 2017); and **3)** perturbation methods that shuffle or modify the communicated messages (e.g., Abadi et al., 2016; McMahan et al., 2018; Erlingsson et al., 2019; Cheu et al., 2019; Zhu et al., 2019). Our protection techniques belong to the third category, as we add random perturbations to the gradients to protect the labels. Many randomness-based protection methods have been proposed in the domain of horizontal FL. In this case, *differential privacy* (DP) (Dwork, 2006; Dwork et al., 2014) is commonly used to measure the proposed random mechanisms' ability to anonymize the identity of any single participating example in the model iterates. However, in split learning, after PSI, both parties know *exactly the identity of which example has participated in a given gradient update*. As we explain in Section 3.1, the object we aim to protect (the communicated cut layer gradients), unlike the model iterates, is not an aggregate function of all the examples but are instead example-specific. **As a result, DP and its variants** (e.g. label DP (Chaudhuri & Hsu, 2011; Ghazi et al., 2021)) **are not directly applicable metrics in our setting, and we instead propose a different metric** (discussed in Section 3.2).

## 3  LABEL LEAKAGE IN SPLIT LEARNING

We first introduce the two-party split learning problem for binary classification, and then formally describe our threat model and privacy quantification metrics with two concrete attack examples.

### 3.1  TWO-PARTY SPLIT LEARNING IN BINARY CLASSIFICATION

**Problem setup.** Consider two parties learning a composition model $h \circ f$ jointly for a binary classification problem over the domain $\mathcal{X} \times \{0, 1\}$ (Figure 1). The non-label party owns the representation function $f : \mathcal{X} \to \mathbb{R}^d$ and each example's raw feature $X \in \mathcal{X}$ while the label party owns the logit function $h : \mathbb{R}^d \to \mathbb{R}$ and each example's label $y \in \{0, 1\}$[2]. Let $\ell = h(f(X))$ be the logit of the positive class whose predicted probability is given through the sigmoid function: $\widetilde{p}_1 = 1/(1 + \exp(-\ell))$. We measure the loss of such prediction through the cross entropy loss $L = \log(1 + \exp(-\ell)) + (1 - y)\ell$. During model inference, the non-label party computes $f(X)$ and sends it to the label party who will then execute the rest of forward computation in Figure 1.

**Model training** (Figure 1: backward gradient computation). To train the model using gradient descent, the label party starts by first computing the gradient of the loss $L$ with respect to the logit $\frac{dL}{d\ell} = (\widetilde{p}_1 - y)$. Using the chain rule, the label party can then compute the gradient of $L$ with respect to its function $h$'s parameters and perform the gradient updates. To also allow the non-label party to learn its function $f$, the label party needs to additionally compute the gradient with respect to cut layer feature $f(X)$ and communicate it to the non-label party. We denote this gradient by $g \coloneqq \nabla_{f(X)} L = (\widetilde{p}_1 - y)\nabla_z h(z)|_{z=f(X)} \in \mathbb{R}^d$ (by chain rule). After receiving $g$, the non-label party continues the backpropagation towards $f$'s parameters and also perform the gradient updates.

**Why Not Differential Privacy?** Note that for a given iteration, the non-label party randomly chooses $B$ example IDs to form a batch. Therefore, the identity of which examples are used is known to the non-label party by default. In addition, the communicated features $f(X)$ and returned gradients $g$ will both be matrices in $\mathbb{R}^{B \times d}$ with each row belonging to a specific example in the batch. The different gradients (rows of the matrix) are not with respect to the same model parameters, but are instead with respect to different examples' cut-layer features; thus, no averaging over or shuffling of the rows of the gradient matrix can be done prior to communication to ensure correct computation of $f$'s parameters on the non-label party side. *This example-aware and example-specific nature of the communicated gradient matrix makes differential privacy* (which focuses on anonymizing an example's participation in an aggregate function) *inapplicable for this problem* (see also Section 2).

### 3.2  THREAT MODEL AND PRIVACY QUANTIFICATION

Below we specify several key aspects of our threat model, including the adversary's objective and capabilities, our metric for quantifying privacy loss, and the possible inclusion of side information.

**Adversary's objective.** At a given moment in time during training (with $f$ and $h$ fixed), since the communicated cut layer gradient $g$ is a **deterministic function** of $y$ (see Section 3.1), we consider an adversarial non-label party whose objective is to recover the label party's hidden label $y$ based on the information contained in $g$ for every training example.

**Adversary's capability.** We consider an *honest-but-curious* non-label party which cannot tamper with training by selecting which examples to include in a batch or sending incorrect features $f(X)$; instead, we assume that the adversary follows the agreed-upon split training procedure while trying to guess the label $y$. This can be viewed as a binary classification problem where the (input, output) distribution is the induced distribution of $(g, y)$. We allow the adversary to use any binary classifier $q : \mathbb{R}^d \to \{0, 1\}$ to guess the labels. This classifier can be represented by a (scoring function $r$, threshold $t$) tuple, where $r : \mathbb{R}^d \to \mathbb{R}$ maps an example's cut layer gradient to a real-valued score and the threshold $t \in \mathbb{R}$ determines a cut-off so that $q(g) = 1$ if $r(g) > t$ and $q(g) = 0$ if $r(g) \leq t$. Moving forward, we use this tuple representation to describe adversarial non-label party classifiers.

**Privacy loss quantification.** As we consider binary classification, a natural metric to quantify the performance of an adverary's scoring function $r$ is the AUC of its ROC curve. Denote the unperturbed class-conditional distributions of the cut-layer gradients by $P^{(1)}$ and $P^{(0)}$ for the

---

[2]To simplify notation, we assume no additional features in the label party to compute the logit. The data leakage problem still holds true for other more complicated settings (see WDL experiment setting in Section 5).

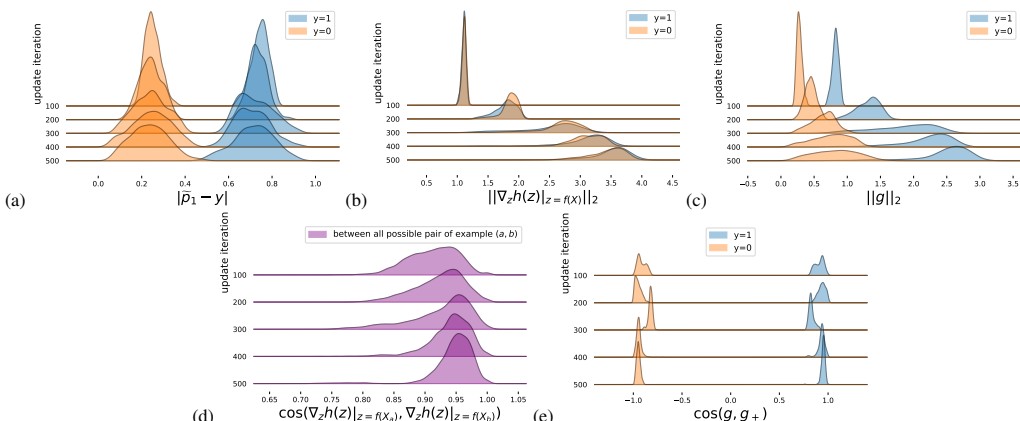

Figure 2: Distributions of quantities discussed in Observations 1-4 after the first 100, 200, 300, 400, 500 steps of stochastic gradient descent training of the WDL model on Criteo (see experiments).

positive and negative class, respectively. The ROC curve of a scoring function $r$ is a parametric curve $t \mapsto (\mathrm{FPR}_r(t), \mathrm{TPR}_r(t)) \in [0,1]^2$ which maps a threshold value $t \in \mathbb{R}$ to the corresponding (*False Positive Rate*, *True Positive Rate*) tuple of the classifier represented by $(r, t)$, with $\mathrm{FPR}_r(t) \coloneqq P^{(0)}(\{g : r(g) > t\})$ and $\mathrm{TPR}_r(t) \coloneqq P^{(1)}(\{g : r(g) > t\})$. The AUC of the ROC curve of a scoring function $r$ (denote by $\mathrm{AUC}(r)$) can be expressed as an integral:

$$\mathrm{AUC}(r) = \int_{\infty}^{-\infty} \mathrm{TPR}_r(t) \; d\mathrm{FPR}_r(t) \quad \in [0,1] \qquad \textbf{(Leak AUC)}$$

(more details on this expression see Appendix A.1.) We use this value as the privacy loss quantification metric for a specific adversary scoring function $r$ and refer to it as the *leak AUC*. This metric summarizes the predictive performance of all classifiers that can be constructed through all threshold values $t$ and removes the need to tune this classifier-specific hyperparameter. The leak AUC being close to $1$ implies that the corresponding scoring function $r$ can very accurately recover the private label, whereas a value of around $0.5$ means $r$ is non-informative in predicting the labels. In practice, during batch training, the leak AUC of $r$ can be estimated at every gradient update iteration using the minibatch of cut-layer gradients together with their labels.

**Side information.** Among all the scoring functions within our threat model, it is conceivable that only some would recover the hidden labels accurately. Picking such effective ones would require the non-label party to have population-level side information specifically regarding the properties of (and distinction between) the positive and negative class's cut-layer gradient distributions. Since we allow the adversary to pick any specific (measurable) scoring function, we implicitly allow for such population-level side information for the adversary. However, we assume the non-label party has *no example-level side information* that is different example by example. Thus we also don't use local DP for privacy quantification (detailed explanation in Appendix A.8). Next we provide two example scoring functions which use population-level side-information to effectively recover the label.

### 3.3 PRACTICAL ATTACK METHODS

**Attack 1: Norm-based scoring function.** Note that $\|g\|_2 = |\widetilde{p}_1 - y| \cdot \|\nabla_a h(a)|_{a=\boldsymbol{f}(X)}\|_2$. We make the following observations for $|\widetilde{p}_1 - y|$ and $\|\nabla_a h(a)|_{a=\boldsymbol{f}(X)}\|_2$, which hold true for a wide range of real-world learning problems.

- **Observation 1:** Throughout training, the model tends to be *less confident about a positive example being positive than a negative example being negative.* In other words, the confidence gap of a positive example $1 - \widetilde{p}_1 = |\widetilde{p}_1 - y|$ (when $y = 1$) is typically larger than the confidence gap of a negative example $1 - \widetilde{p}_0 = \widetilde{p}_1 = |\widetilde{p}_1 - y|$ (when $y = 0$) (see Figure 2(a)). This observation is particularly true for problems like advertising conversion prediction and disease prediction, where there is inherently more ambiguity for the positive class than the negative. For example, in advertising, uninterested users of a product will never click on its ad and convert, but those interested, even after clicking, might make the purchase only a fraction of the time depending on time/money constraints. (See A.2 for such an ambiguity even for a class-balanced setting.)

- **Observation 2**: Throughout training, the norm of the gradient vector $\|\nabla_z h(z)|_{z=\boldsymbol{f}(X)}\|_2$ is on the same order of magnitude (has similar distribution) for both the positive and negative examples (Figure 2(b)). This is natural because $\nabla_a h(a)|_{a=\boldsymbol{f}(X)}$ is not a function of $y$.

As a consequence of Observation 1 and 2, *the gradient norm $\|g\|_2$ of the positive instances are generally larger than that of the negative ones* (Figure 2(c)). Thus, the scoring function $r_n(g) = \|g\|_2$ is a strong predictor of the unseen label $y$. We name the privacy loss (leak AUC) measured against the attack $r_n$ the ***norm leak AUC***. In Figure 2(c), the norm leak AUCs are consistently above 0.9, signaling a high level of label leakage throughout training.

**Attack 2: Direction-based scoring function.** We now show that the direction of $g$ (in addition to its magnitude) can also leak the label. For a pair of examples, $(X_a, y_a), (X_b, y_b)$, let their respective predicted positive class probability be $\widetilde{p}_{1,a}, \widetilde{p}_{1,b}$ and their communicated gradients be $g_a$, $g_b$. Let $\cos : \mathbb{R}^d \times \mathbb{R}^d \to \mathbb{R}$ denote the cosine similarity function $\cos(g_a, g_b) = g_a^T g_b / (\|g_a\|_2 \|g_b\|_2)$. It is easy to see that $\cos(g_a, g_b) = \text{sgn}(\widetilde{p}_{1,a} - y_a) \cdot \text{sgn}(\widetilde{p}_{1,a} - y_b) \cdot \cos(\nabla_z h(z)|_{z=\boldsymbol{f}(X_a)}, \nabla_z h(z)|_{z=\boldsymbol{f}(X_b)})$, where sgn(x) is the sign function which returns 1 if $x \geq 0$, and $-1$ if $x < 0$. We highlight two additional observations that can allow us to use cosine similarity to recover the label.

- **Observation 3**: When the examples $a, b$ are of different classes, the term $\text{sgn}(\widetilde{p}_{1,a} - y_a) \cdot \text{sgn}(\widetilde{p}_{1,a} - y_b) = -1$ is negative. On the other hand, when examples $a$, $b$ are of the same class (both positive/both negative), this product will have a value of 1 and thus be positive.

- **Observation 4**: Throughout training, for any two examples $a, b$, their gradients of the function $h$ always form an acute angle, *i.e.* $\cos(\nabla_z h(z)|_{z=\boldsymbol{f}(X_a)}, \nabla_z h(z)|_{z=\boldsymbol{f}(X_b)}) > 0$ (Figure 2(d)). For neural networks that use monotonically increasing activation functions (such as ReLU, sigmoid, $\tanh$), this is caused by the fact that the gradients of these activation functions with respect to its inputs are coordinatewise nonnegative and thus always lie in the first closed hyperorthant.

Since $\cos(g_a, g_b)$ is the product of the terms from Observation 3 and 4, we see that *for a given example, all the examples that are of the same class result in a positive cosine similarity, while all opposite class examples result in a negative cosine similarity.* If the problem is class-imbalanced and the non-label party knows there are fewer positive examples than negative ones, it can thus determine the label of each example: the class is negative if more than half of the examples result in positive cosine similarity; otherwise it is positive. For many practical applications, *the non-label party may reasonably guess which class has more examples in the dataset a priori without ever seeing any data*—for example, in disease prediction, the percentage of the entire population having a certain disease is almost always much lower than 50%; in online advertising conversion prediction, the conversion rate (fraction of positive examples) is rarely higher than 30%. Note that the *non-label party doesn't need knowledge of the exact sample proportion of each class* for this method to work.

To simplify this attack for evaluation, we consider an even worse oracle scenario where the non-label party knows the clean gradient of one positive example $g_+$. Unlike the aforementioned practical majority counting attack which needs to first figure out the direction of one positive gradient, this oracle scenario assumes the non-label party is directly given this information. Thus, any protection method capable of defending this oracle attack would also protect against the more practical one. With $g_+$ given, the direction-based scoring function $r_d$ is simply $r_d(g) = \cos(g, g_+)$. We name the privacy loss (leak AUC) against this oracle attack $r_d$ the ***cosine leak AUC***. In practice, we randomly choose a positive class clean gradient from each batch as $g_+$ for evaluation. For iterations in Figure 2(e), the cosine leak AUC all have the highest value of 1 (complete label leakage).

## 4 LABEL LEAKAGE PROTECTION METHODS

In this section, we first introduce a heuristic random perturbation approach designed to prevent the practical attacks identified in Section 3.3. We then propose a theoretically justified method that aims to protect against the *entire class of scoring functions* considered in our threat model (Section 3.2).

### 4.1 A HEURISTIC PROTECTION APPROACH

**Random perturbation and the isotropic Gaussian baseline.** To protect against label leakage, the label party should ideally communicate essential information about the gradient without communicating its actual value. Random perturbation methods generally aim to achieve this goal. One obvious consideration for random perturbation is to keep the perturbed gradients unbiased. In other words, suppose $\tilde{g}$ is the perturbed version of an example's true gradient $g$, then we want $\mathbb{E}[\tilde{g} \mid g] = g$. By chain rule and linearity of expectation, this ensures the computed gradients of the non-label party's parameters $f$ will also be unbiased, a desirable property for stochastic optimization. Among unbiased perturbation methods, a simple approach is to add *iid* isotropic Gaussian noise to every gradient to mix the positive and negative gradient distribution before sending to the non-label party.

Although isotropic Gaussian noise is a valid option, it may not be optimal because **1)** the gradients are vectors but not scalars, so the structure of the noise covariance matrix matters. Isotropic noise might neglect the direction information; **2)** due to the asymmetry of the positive and negative gradient distribution, the label party could add noise with different distributions to each class's gradients.

**Norm-alignment heuristic.** We now introduce an improved heuristic approach of adding zero-mean Gaussian noise with non-isotropic and example-dependent covariance. [**Magnitude choice**] As we have seen that $\|g\|_2$ can be different for positive and negative examples and thus leak label information, this heuristic first aims to make the norm of each perturbed gradient indistinguishable from one another. Specifically, we want to match the expected squared 2-norm of every perturbed gradient in a mini-batch to the largest squared 2-norm in this batch (denote by $\|g_{\max}\|_2^2$). [**Direction choice**] In addition, as we have seen empirically from Figure 2(e), the positive and negative gradients lie close to a one-dimensional line in $\mathbb{R}^d$, with positive examples pointing in one direction and negative examples in the other. Thus we consider only adding noise (roughly speaking) along "this line". More concretely, for a gradient $g_j$ in the batch, we add a zero-mean Gaussian noise vector $\eta_j$ supported only on the one-dimensional space along the line of $g_j$. In other words, the noise's covariance is the rank-1 matrix $\text{Cov}[\eta_j] = \sigma_j^2 g_j g_j^T$. To calculate $\sigma_j$, we aim to match $\mathbb{E}[\|g_j + \eta_j\|_2^2] = \|g_{\max}\|_2^2$. A simple calculation gives $\sigma_j = \sqrt{\|g_{\max}\|_2^2/\|g_j\|_2^2 - 1}$. Since we align to the maximum norm, we name this heuristic protection method `max_norm`. The advantage of `max_norm` is that it has no parameter to tune. Unfortunately, it does not have a strong theoretical motivation, cannot flexibly trade-off between model utility and privacy, and may be broken by some unknown attacks.

## 4.2 OPTIMIZED PERTURBATION METHOD: MARVELL

Motivated by the above issues of `max_norm`, we next study how to achieve a more principled trade-off between model performance (utility) and label protection (privacy). To do so, we directly minimize the worst-case adversarial scoring function's leak AUC under a utility constraint. We name this protection method `Marvell` (opti**M**ized perturb**A**tion to p**R**e**VE**nt **L**abel **L**eakage).

**Noise perturbation structure.** Due to the distribution difference between the positive and negative class's cut layer gradients, we consider having the label party additively perturb the randomly sampled positive $g^{(1)}$ and negative $g^{(0)}$ gradients with independent zero-mean random noise vectors $\eta^{(1)}$ and $\eta^{(0)}$ with possibly different distributions (denoted by $D^{(1)}$ and $D^{(0)}$). We use $\widetilde{P}^{(1)}$ and $\widetilde{P}^{(0)}$ to denote the induced perturbed positive and negative gradient distributions. Our goal is to find the optimal noise distributions $D^{(1)}$ and $D^{(0)}$ by optimizing our privacy objective described below.

**Privacy protection optimization objective.** As the adversarial non-label party in our threat model is allowed to use any measurable scoring function $r$ for label recovery, we aim to protect against *all such scoring functions* by minimizing the privacy loss of the worst case scoring function measured through our leak AUC metric. Formally, our optimization objective is $\min_{D^{(1)}, D^{(0)}} \max_r \text{AUC}(r)$. Here to compute $\text{AUC}(r)$, the $\text{FPR}_r(t)$ and $\text{TPR}_r(t)$ needs to be computed using the perturbed distributions $\widetilde{P}^{(1)}$ and $\widetilde{P}^{(0)}$ instead of the unperturbed $P^{(1)}$ and $P^{(0)}$ (Section 3.2). Since AUC is difficult to directly optimize, we consider optimizing an upper bound through the following theorem:

**Theorem 1.** *For $0 \le \epsilon < 4$ and any perturbed gradient distributions $\widetilde{P}^{(1)}$ and $\widetilde{P}^{(0)}$ that are absolutely continuous with respect to each other,*

$$\text{KL}(\widetilde{P}^{(1)} \parallel \widetilde{P}^{(0)}) + \text{KL}(\widetilde{P}^{(0)} \parallel \widetilde{P}^{(1)}) \le \epsilon \quad \textit{implies} \quad \max_r \text{AUC}(r) \le \tfrac{1}{2} + \tfrac{\sqrt{\epsilon}}{2} - \tfrac{\epsilon}{8}.$$

From Theorem 1 (proof in Appendix A.3), we see that as long as the sum KL divergence is below 4, the smaller sumKL is, the smaller $\max_r \text{AUC}(r)$ is. ($1/2 + \sqrt{\epsilon}/2 - \epsilon/8$ decreases as $\epsilon$ decreases.) Thus we can instead minimize the sum KL divergence between the perturbed gradient distributions:

$$\text{sumKL}^* := \min_{D^{(1)}, D^{(0)}} \text{KL}(\widetilde{P}^{(1)} \parallel \widetilde{P}^{(0)}) + \text{KL}(\widetilde{P}^{(0)} \parallel \widetilde{P}^{(1)}). \tag{1}$$

**Utility constraint.** In an extreme case, we could add infinite noise to both the negative and positive gradients. This would minimize (1) optimally to 0 and make the worst case leak AUC 0.5, which is equivalent to a random guess. However, stochastic gradient descent cannot converge under infinitely large noise, so it is necessary to control the variance of the added noise. We thus introduce the noise power constraint: $p \cdot \text{tr}(\text{Cov}[\eta^{(1)}]) + (1 - p) \cdot \text{tr}(\text{Cov}[\eta^{(0)}]) \le P$, where $p$ is the fraction of positive examples (already known to the label party); $\text{tr}(\text{Cov}[\eta^{(i)}])$ denotes the trace of the covariance matrix of the random noise $\eta^{(i)}$; and the upper bound $P$ is a tunable hyperparameter to control the level of noise: larger $P$ would achieve a lower sumKL and thus lower worst-case leak AUC and better **privacy**; however, it would also add more noise to the gradients, leading to slower optimization

convergence and possibly worse model **utility**. We weight each class's noise level $\mathrm{tr}(\mathrm{Cov}[\eta^{(i)}])$ by its example proportion ($p$ or $1-p$) since, from an optimization perspective, we want to equally control every training example's gradient noise. The constrained optimization problem becomes:

$$\min_{D^{(1)}, D^{(0)}} \mathrm{KL}(\widetilde{P}^{(1)} \parallel \widetilde{P}^{(0)}) + \mathrm{KL}(\widetilde{P}^{(0)} \parallel \widetilde{P}^{(1)}) \quad \text{s.t.} \quad p \cdot \mathrm{tr}(\mathrm{Cov}[\eta^{(1)}]) + (1-p) \cdot \mathrm{tr}(\mathrm{Cov}[\eta^{(0)}]) \leq P. \quad (2)$$

**Optimizing the objective in practice.** To solve the optimization problem we first introduce some modelling assumptions. We assume that the unperturbed gradient of each class follows a Gaussian distribution: $g^{(1)} \sim \mathcal{N}(\bar{g}^{(1)}, vI_{d \times d})$ and $g^{(0)} \sim \mathcal{N}(\bar{g}^{(0)}, uI_{d \times d})$. Despite this being an approximation, as we see later in Section 5, it can achieve strong protection quality against our identified attacks. In addition, it makes the optimization easier (see below) and provides us with insight on the optimal noise structure. We also search for perturbation distributions that are Gaussian: $D^{(1)} = \mathcal{N}(0, \Sigma_1)$ and $D^{(0)} = \mathcal{N}(0, \Sigma_0)$ with commuting covariance matrices: $\Sigma_1 \Sigma_0 = \Sigma_0 \Sigma_1$. The commutative requirement slightly restricts our search space but also makes the optimization problem more tractable. Our goal is to solve for the optimal noise structure, *i.e.* the positive semidefinite covariance matrices $\Sigma_0$, $\Sigma_1$. Let $\Delta g := \bar{g}^{(1)} - \bar{g}^{(0)}$ denote the difference between the positive and negative gradient's mean vectors. We now have the following theorem (proof and interpretation in Appendix A.4):

**Theorem 2.** *The optimal $\Sigma_1^*$ and $\Sigma_0^*$ to (2) with the above assumptions have the form:*

$$\Sigma_1^* = \frac{\lambda_1^{(1)*} - \lambda_2^{(1)*}}{\|\Delta g\|_2^2}(\Delta g)(\Delta g)^\top + \lambda_2^{(1)*}I_d, \quad \Sigma_0^* = \frac{\lambda_1^{(0)*} - \lambda_2^{(0)*}}{\|\Delta g\|_2^2}(\Delta g)(\Delta g)^\top + \lambda_2^{(0)*}I_d, \quad (3)$$

*where $(\lambda_1^{(0)*}, \lambda_2^{(0)*}, \lambda_1^{(1)*}, \lambda_2^{(1)*})$ is the solution to the following 4-variable optimization problem:*

$$\min_{\lambda_1^{(0)}, \lambda_1^{(1)}, \lambda_2^{(0)}, \lambda_2^{(1)}} (d-1)\frac{\lambda_2^{(0)} + u}{\lambda_2^{(1)} + v} + (d-1)\frac{\lambda_2^{(1)} + v}{\lambda_2^{(0)} + u} + \frac{\lambda_1^{(0)} + u + \|\Delta g\|_2^2}{\lambda_1^{(1)} + v} + \frac{\lambda_1^{(1)} + v + \|\Delta g\|_2^2}{\lambda_1^{(0)} + u}$$

$$\text{s.t.} \quad p\lambda_1^{(1)} + p(d-1)\lambda_2^{(1)} + (1-p)\lambda_1^{(0)} + (1-p)(d-1)\lambda_2^{(0)} \leq P,$$

$$-\lambda_1^{(1)} \leq 0, \quad -\lambda_1^{(0)} \leq 0, \quad -\lambda_2^{(1)} \leq 0, \quad -\lambda_2^{(0)} \leq 0, \quad \lambda_2^{(1)} - \lambda_1^{(1)} \leq 0, \quad \lambda_2^{(0)} - \lambda_1^{(0)} \leq 0$$

**Additional details of** `Marvell`**.** By Theorem 2, our optimization problem over two positive semidefinite matrices is reduced to a much simpler 4-variable optimization problem. We include a detailed description of how the constants in the problem are estimated in practice and what solver we use in a full description of the `Marvell` algorithm in Appendix A.5. Beyond optimization details, it is worth noting how to set the power constraint hyperparameter $P$ in Equation 2 in practice. As directly choosing $P$ requires knowledge of the scale of the gradients in the specific application and the scale could also shrink as the optimization converges, we instead express $P = s \|\Delta g\|_2^2$, and tune for a fixed hyperparameter $s > 0$. This alleviates the need to know the scale of the gradients in advance, and the resulting value of $P$ can also dynamically change throughout training as the distance between the two gradient distributions' mean $\|\Delta g\|_2$ changes.

## 5 EXPERIMENTS

In this section, we first describe our experiment setup and then demonstrate the label protection quality of `Marvell` as well as its privacy-utility trade-off relative to baseline approaches.

**Empirical Setup.** We use three real-world binary classification datasets for evaluation: Criteo and Avazu, two online advertising prediction datasets with millions of examples; and ISIC, a healthcare image dataset for skin cancer prediction. All three datasets exhibit severe label leakage problem without protection. (see Appendix A.6.1 on dataset and preprocessing details). We defer similar results on Avazu to Appendix A.7 and focus on Criteo and ISIC in this section. For Criteo, we train a Wide&Deep model (Cheng et al., 2016) where the non-label party owns the embedding layers for input features and the first three 128-unit ReLU activated MLP layers (first half of the deep part) while the label party owns the remaining layers of the deep part and the entire wide part of the model[3]. For ISIC, we train a model with 6 convolutional layers each with 64 channels followed by a 64-unit ReLU MLP layer, and the cut layer is after the fourth convolutional layer. In this case, an example's cut layer feature $f(X)$ and gradient $g$ are both in $\mathbb{R}^{5 \times 5 \times 64}$. We treat such tensors as vectors in $\mathbb{R}^{1600}$ to fit into our analysis framework (for additional model architecture and training details see Appendix A.6.2, A.6.3).

---

[3]In this setting, the label party will also process input features (through the wide part) just like the non-label party, further relaxing our formal split learning setup in Section 3.

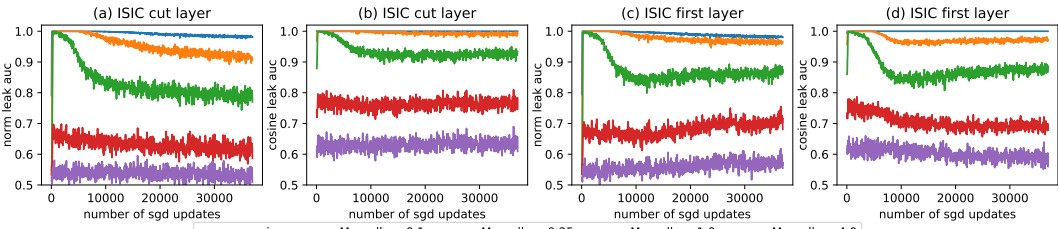

Figure 3: Norm and cosine leak AUC (computed every batch) at the cut layer and at the first layer under no protection vs. `Marvell` with different scale hyperparameter $s$ throughout the ISIC training.

## 5.1 LABEL LEAKAGE AND MARVELL'S STRONG AND FLEXIBLE PROTECTION

We first evaluate the protection quality of `Marvell` against the norm and cosine attacks discussed in Section 3.3. We also compare against the leakage metrics when no protection is applied (`no_noise`). As the results across the three datasets are highly similar, we use ISIC as an example (other datasets see Appendix A.7.1). We see in Figure 3(a)(b) that unlike `no_noise` where the label information is completely leaked (leak AUC $\approx 1$) throughout training, `Marvell` **achieves a flexible degree of protection (by varying $s$) against both the norm 2(a) and direction attacks 2(b) on the cut layer gradients and has strong protection (leak AUC $\approx 0.5$) at $s = 4.0$.** Additionally, it is natural to ask *whether the gradients of layers before the cut layer (on the non-label party side) can also leak the labels* as the non-label party keeps back propagating towards the first layer. In Figure 3(c)(d), we compute the leak AUC values when using the non-label party's first layer activation gradient as inputs to the scoring functions to predict $y$. Without protection, the first layer gradient still leaks the label very consistently. In contrast, `Marvell` still achieves strong privacy protection at the first layer ($s = 4.0$) despite the protection being analyzed at the cut layer.

## 5.2 PRIVACY-UTILITY TRADE-OFF COMPARISON

After showing `Marvell` can provide strong privacy protection against our identified attacks, we now see how well it can preserve utility by comparing its privacy-utility tradeoff against other protection baselines: `no_noise`, isotropic Gaussian (`iso`), and our proposed heuristic `max_norm`. Similar to how we allow `Marvell` to use a power constraint to depend on the current iteration's gradient distribution through $P = s\|\Delta g\|_2^2$, we also allow `iso` to have such type of dependence— specifically, we add $\eta \sim \mathcal{N}(\mathbf{0}, (t/d) \cdot \|g_{\max}\|_2^2 I_{d \times d})$ to every gradient in a batch with $t$ a tunable privacy hyperparameter to be fixed throughout training. To trace out the complete tradeoff curve for `Marvell` and `iso`, we conduct more than 20 training runs for each protection method with a different value of privacy hyperparameter ($s$ for `Marvell`, $t$ for `iso`) in each run on every dataset. (Note that `no_noise` and `max_norm` do not have privacy hyperparameters.)

We present the tradeoffs between privacy (measured through norm and cosine leak AUC at cut layer/first layer) and utility (measured using test loss and test AUC) in Figure 4. To summarize the leak AUC over a given training run, we pick the 95% quantile over the batch-computed leak AUCs throughout all training iterations. This quantile is chosen instead of the mean because we want to measure the most-leaked iteration's privacy leakage (highest leak AUC across iterations) to ensure the labels are not leaked at any points during training. 95% quantile is chosen instead of the max (100%) as we want this privacy leak estimate to be robust against randomness of the training process.

**Privacy-Utility Tradeoff comparison results.** In measuring the privacy-utility tradeoff, we aim to find a method that consistently achieves a lower leak AUC (better privacy) for the same utility value.

- **[Marvell vs `iso`]** As shown in Figure 4, `Marvell` *almost always achieves a better tradeoff than* `iso` *against both of our proposed attacks at both the cut layer and the first layer on both the* ISIC *and* Criteo *datasets.* It is important to note that although the utility constraint is in terms of training loss optimization, `Marvell`'s better tradeoff still translates to the generalization performance when the utility is measured through test loss or test AUC. Additionally, despite achieving reasonable (though still worse than `Marvell`) privacy-utility tradeoff against the norm-based attack, `iso` performs much worse against the direction-based attack: on ISIC, even after applying a significant amount of isotropic noise (with $t > 20$), `iso`'s cosine leak AUC is still higher than $0.9$ at the cut layer (Figure 4(b,f)). In contrast, `Marvell` is effective against this direction-based attack with a much lower cosine leak AUC $\approx 0.6$.

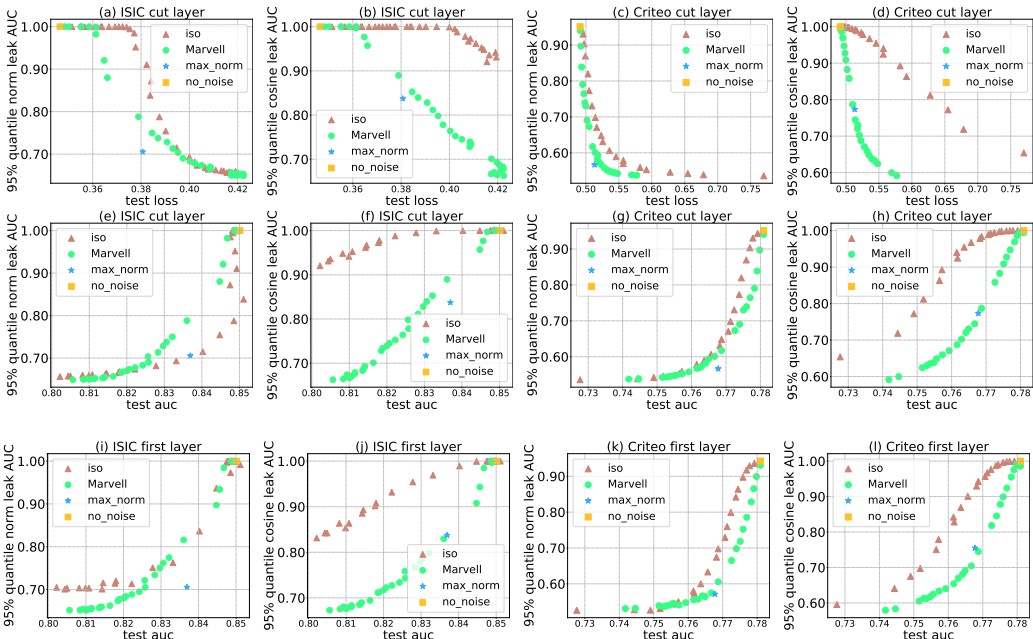

Figure 4: Privacy (norm & cosine leak AUC) vs Utility (test loss & test AUC) trade-off of protection methods (Marvell, iso, no_noise, max_norm) at the cut and first layer on ISIC and Criteo.

- [max_norm **heuristic**] Beyond Marvell, we see that our heuristic approach max_norm can match and sometimes achieve even lower (Figure 4(a,f,i)) leak AUC value than Marvell at the same utility level. We believe this specifically results from our norm and direction consideration when designing this heuristic. However, without a tunable hyperparameter, max_norm cannot tradeoff between privacy and utility. Additionally, unlike Marvell which is designed to protect against the entire class of adversarial scoring functions, max_norm might still fail to protect against other future attack methods beyond those considered here.

In summary, our principled method Marvell **significantly outperforms the isotropic Gaussian baseline**, and our proposed max_norm **heuristic can also work particularly well against the norm- and direction-based attacks** which we identified in Section 3.3.

## 6   CONCLUSION

In this paper, we formulate a label leakage threat model in the two-party split learning binary classification problem through a novel privacy loss quantification metric (leak AUC). Within this threat model, we provide two simple yet effective attack methods that can accurately uncover the private labels of the label party. To counter such attacks, we propose a heuristic random perturbation method max_norm as well as a theoretically principled method Marvell which searches for the optimal noise distributions to protect against the worst-case adversaries in the threat model. We have conducted extensive experiments to demonstrate the effectiveness of Marvell and max_norm over the isotropic Gaussian perturbation baseline iso.

**Open questions and future work.** Our work is the first we are aware of to identify, rigorously quantify, and protect against the threat of label leakage in split-learning, and opens up a number of worthy directions of future study. In particular, as the model parameters are updated every batch in our problem setup, the true gradient of an example and the gradient distribution would both change. An interesting question is whether the adversarial non-label party can remember the stale gradient of the same example from past updates (possibly separated by hundreds of updates steps) in order to recover the label information in the current iteration in a more complex threat model. It would also be interesting to build on our results to study whether there exist attack methods when the classification problem is multiclass instead of binary, and when the split learning scenario involves more than two parties with possibly more complicated training communication protocols (e.g., Vepakomma et al., 2018).

**Ethics Statement.** In our paper, we have identified a realistic threat of label leakage in the two-party split learning binary classification problem. We aim to raise awareness about potential privacy issues in this problem domain, where many industrial applications have been deployed. Beyond making such threats clear, we have taken the first steps towards protection—we have proposed both heuristic and principled methods that can preserve label privacy. We hope our work will pave the way for future analyses that make the two-party split learning framework more effective and secure.

**Reproducibility Statement.** To make our paper reproducible, we provide:

- Proofs of our Theorem 1 and Theorem 2 in Appendix A.3,A.4;
- Detailed experiment description including 1) data preprocessing, 2) model architecture, 3) training algorithm and hyperparameters in Appendix A.6.
- Source code with running instructions (in README.md) at
  https://github.com/OscarcarLi/label-protection.

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

# A APPENDIX

**Appendix Outline**

A.1   EXPRESSING AUC($r$) AS AN INTEGRAL

Recall that the ROC curve of a scoring function $r : \mathbb{R}^d \to \mathbb{R}$ is a parametric curve $c_r : \mathbb{R} \to [0,1]^2$ such that $c_r(t) = (\text{FPR}_r(t), \text{TPR}_r(t))$, with

$$\text{FPR}_r : \mathbb{R} \to [0,1], \text{ such that } \text{FPR}_r(t) := P^{(0)}\left(\left\{g \in \mathbb{R}^d : r(g) > t\right\}\right)$$

$$\text{TPR}_r : \mathbb{R} \to [0,1], \text{ such that } \text{TPR}_r(t) := P^{(1)}\left(\left\{g \in \mathbb{R}^d : r(g) > t\right\}\right).$$

We notice that $\text{FPR}_r$ and $\text{TPR}_r$ are both monotonically decreasing functions in $t$ with

$$\lim_{t \to -\infty} \text{FPR}_r(t) = 1, \quad \lim_{t \to \infty} \text{FPR}_r(t) = 0$$

$$\lim_{t \to -\infty} \text{TPR}_r(t) = 1, \quad \lim_{t \to \infty} \text{TPR}_r(t) = 0.$$

However, $\text{FPR}_r$ and $\text{TPR}_r$ does not need to be differentiable everywhere with respect to $t$. Thus we cannot simply express the area under the curve using $\int_{\infty}^{-\infty} \text{TPR}_r(t)\, \text{FPR}'_r(t) dt$. On the other hand, because both $\text{FPR}_r$ and $\text{TPR}_r$ are functions of bounded variations, we can express the area under the parametric curve $c_r$ through the Riemann-Stieltjes integral which involves integrating the function $\text{TPR}_r$ with respect to the function $\text{FPR}_r$:

$$\text{AUC}(r) = \int_{t=\infty}^{t=-\infty} \text{TPR}_r(t)\; d\text{FPR}_r(t)$$

Here the integration boundary is from $\infty$ to $-\infty$ in order to ensure the integral evaluates to a positive value.

A.2   TOY CLASS-BALANCED EXAMPLE OF POSITIVE EXAMPLE PREDICTION LACKING CONFIDENCE

As mentioned in Observation 1 of norm-based attack in Section 3.3, for many practical applications, there is inherently more ambiguity for the positive class than the negative class. To make this more concrete, let's consider a simple toy example.

Suppose the binary classification problem is over the real line. Here, 50% of the data is positive and lies uniformly in the interval $[0,1]$. On the other hand, the remaining negative data (the other 50% of the total data) is a mixture distribution: 10% of the negative data also lies uniformly in $[0,1]$, while the rest 90% of the negative data lies uniformly in $[1,2]$.

$$p(x \mid y = 1) = \mathbb{1}(x \in [0,1]), \qquad\qquad\qquad p(y = 1) = 0.5.$$
$$p(x \mid y = 0) = 0.1 \cdot \mathbb{1}(x \in [0,1]) + 0.9 \cdot \mathbb{1}(x \in [1,2]), \qquad p(y = 0) = 0.5.$$

This setup mirrors our online advertising example where for all the users interested in the product (with feature $x \in [0,1]$), only a part of them would actually make the purchase after clicking on its advertisement.

In this case, the best possible probabilistic classifiers $C^{\text{opt}}$ that can be ever learned, by Bayes Rule, would predict any example in $[0,1]$ to be of the positive class with probability $\frac{10}{11}$, while it would predict any example in $[1,2]$ to be of the negative class with probability 1:

$$C^{\text{opt}}(y = 1 \mid x) = \begin{cases} \frac{10}{11} & \text{if } x \in [0,1] \\ 0 & \text{if } x \in [1,2] \end{cases}.$$

Thus even for this best possible classifier $C^{\text{opt}}$, every positive example would have a confidence gap of $\frac{1}{11}$ while 90% of the negative examples (the ones in $[1,2]$) would have a confidence gap of 0. Hence we see that in this toy example, our empirical observation of the lack of prediction confidence for positive examples would still hold true.

Besides, it is important to notice that this lack of positive prediction confidence phenomenon happens even in this class-balanced toy example ($p(y = 1) = p(y = 0) = 0.5$). Thus, **Observation 1 does not require the data distribution to be class-imbalanced to hold**, which further demonstrates the generality of our norm-based attack.

### A.3 PROOF OF THEOREM 1

(**Theorem** 1). *For $0 \leq \epsilon < 4$ and any perturbed gradient distributions $\widetilde{P}^{(1)}$ and $\widetilde{P}^{(0)}$ that are absolutely continuous with respect to each other,*

$$\mathrm{KL}(\widetilde{P}^{(1)} \parallel \widetilde{P}^{(0)}) + \mathrm{KL}(\widetilde{P}^{(0)} \parallel \widetilde{P}^{(1)}) \leq \epsilon \quad implies \quad \max_r \mathrm{AUC}(r) \leq \frac{1}{2} + \frac{\sqrt{\epsilon}}{2} - \frac{\epsilon}{8}.$$

*Proof of Theorem 1.* Combining Pinsker's inequality with Jensen's inequality, we can obtain an upper bound of total variation distance by the symmetrized KL divergence (sumKL) for a pair of distributions $(P, Q)$ that are absolutely continuous with respect to each other:

$$\mathrm{TV}(P, Q) \leq \frac{1}{2}(\sqrt{\mathrm{KL}(P \parallel Q)/2} + \sqrt{\mathrm{KL}(Q \parallel P)/2}) \leq \frac{1}{2}\sqrt{\mathrm{KL}(P \parallel Q) + \mathrm{KL}(Q \parallel P)}. \quad (4)$$

By our assumption, this implies that $\mathrm{TV}(\widetilde{P}^{(1)}, \widetilde{P}^{(0)}) \leq \frac{\sqrt{\epsilon}}{2}$. By the equivalent definition of total variation distance $\mathrm{TV}(\widetilde{P}^{(1)}, \widetilde{P}^{(0)})) = \max_{A \subset \mathbb{R}^d}[\widetilde{P}^{(1)}(A) - \widetilde{P}^{(0)}(A)]$, we know that for any $A \subset \mathbb{R}^d$, $\widetilde{P}^{(1)}(A) - \widetilde{P}^{(0)}(A) \leq \frac{\sqrt{\epsilon}}{2}$. For any scoring function $r$ and any threshold value $t$, let $A = \{g : r(g) > t\}$, then we have $\mathrm{TPR}_r(t) - \mathrm{FPR}_r(t) = \widetilde{P}^{(1)}(A) - \widetilde{P}^{(0)}(A) \leq \frac{\sqrt{\epsilon}}{2}$.

Therefore, the AUC of the scoring function $r$ can be upper bounded in the following way:

$$\mathrm{AUC}(r) = \int_{\infty}^{-\infty} \mathrm{TPR}_r(t) \, d\mathrm{FPR}_r(t) \quad (5)$$

$$\leq \int_{\infty}^{-\infty} \min\left(\mathrm{FPR}_r(t) + \frac{\sqrt{\epsilon}}{2}, \ 1\right) d\mathrm{FPR}_r(t), \quad (6)$$

where in (6) we use the additional fact that $\mathrm{TPR}_r(t) \leq 1$ for all $t \in \mathbb{R}$.

When $\epsilon \in [0, 4)$, we have $1 - \frac{\sqrt{\epsilon}}{2} \in (0, 1]$. As $\mathrm{FPR}_r(t)$ is a monotonically nonincreasing function in $t$ with range in $[0, 1]$, the set $\left\{t : \mathrm{FPR}_r(t) \leq 1 - \frac{\sqrt{\epsilon}}{2}\right\} \neq \phi$ is not empty. Let $k := \inf\left\{t : \mathrm{FPR}_r(t) \leq 1 - \frac{\sqrt{\epsilon}}{2}\right\}$. Again by $\mathrm{FPR}_r(t)$ being a monotonically nonincreasing function in $t$, we can break the integration in Equation (6) into two terms:

$$\mathrm{AUC}(r)$$

$$= \int_{\infty}^{k} \min\left(\mathrm{FPR}_r(t) + \frac{\sqrt{\epsilon}}{2}, \ 1\right) d\mathrm{FPR}_r(t) + \int_{k}^{-\infty} \min\left(\mathrm{FPR}_r(t) + \frac{\sqrt{\epsilon}}{2}, \ 1\right) d\mathrm{FPR}_r(t) \quad (7)$$

$$\leq \int_{\infty}^{k} \left(\mathrm{FPR}_r(t) + \frac{\sqrt{\epsilon}}{2}\right) d\mathrm{FPR}_r(t) + \int_{k}^{-\infty} 1 \, d\mathrm{FPR}_r(t) \quad (8)$$

$$= \left[\frac{[\mathrm{FPR}_r(t)]^2}{2} + \frac{\sqrt{\epsilon}}{2}\mathrm{FPR}_r(t)\right]\Bigg|_{t=\infty}^{t=k} + \mathrm{FPR}_r(t)\Bigg|_{t=k}^{t=-\infty} \quad (9)$$

$$\leq \frac{(1 - \frac{\sqrt{\epsilon}}{2})^2}{2} + \frac{\sqrt{\epsilon}}{2}\left(1 - \frac{\sqrt{\epsilon}}{2}\right) + \left[1 - \left(1 - \frac{\sqrt{\epsilon}}{2}\right)\right] \quad (10)$$

$$= \frac{1}{2} + \frac{\sqrt{\epsilon}}{2} - \frac{\epsilon}{8} \quad (11)$$

Since this inequality is true for any scoring function $r$, it is true for the maximum value. Thus the proof is complete. $\square$

## A.4 PROOF AND INTERPRETATION OF THEOREM 2

(**Theorem** 2). *The optimal $\Sigma_1^*$ and $\Sigma_0^*$ to the following problem*

$$\min_{\Sigma_0, \Sigma_1 \in \mathbb{S}} KL(\mathcal{N}(\bar{g}^{(1)}, vI + \Sigma_1) \parallel \mathcal{N}(\bar{g}^{(0)}, uI + \Sigma_0)) + KL(\mathcal{N}(\bar{g}^{(0)}, uI + \Sigma_0) \parallel \mathcal{N}(\bar{g}^{(1)}, vI + \Sigma_1))$$

subject to

$$\Sigma_0 \Sigma_1 = \Sigma_1 \Sigma_0,$$
$$p \cdot \mathrm{tr}(\Sigma_1) + (1 - p) \cdot \mathrm{tr}(\Sigma_0) \leq P,$$
$$\Sigma_1 \succeq \mathbf{0},$$
$$\Sigma_0 \succeq \mathbf{0}.$$

*have the form:*

$$\Sigma_1^* = \frac{\lambda_1^{(1)*} - \lambda_2^{(1)*}}{\|\Delta g\|_2^2}(\Delta g)(\Delta g)^\top + \lambda_2^{(1)*} I_d, \quad \Sigma_0^* = \frac{\lambda_1^{(0)*} - \lambda_2^{(0)*}}{\|\Delta g\|_2^2}(\Delta g)(\Delta g)^\top + \lambda_2^{(0)*} I_d, \quad (12)$$

*where $(\lambda_1^{(0)*}, \lambda_2^{(0)*}, \lambda_1^{(1)*}, \lambda_2^{(1)*})$ is the solution to the following 4-variable optimization problem:*

$$\min_{\lambda_1^{(0)}, \lambda_1^{(1)}, \lambda_2^{(0)}, \lambda_2^{(1)}} (d-1)\frac{\lambda_2^{(0)} + u}{\lambda_2^{(1)} + v} + (d-1)\frac{\lambda_2^{(1)} + v}{\lambda_2^{(0)} + u} + \frac{\lambda_1^{(0)} + u + \|\Delta g\|_2^2}{\lambda_1^{(1)} + v} + \frac{\lambda_1^{(1)} + v + \|\Delta g\|_2^2}{\lambda_1^{(0)} + u}$$

$$s.t. \quad p\lambda_1^{(1)} + p(d-1)\lambda_2^{(1)} + (1-p)\lambda_1^{(0)} + (1-p)(d-1)\lambda_2^{(0)} \leq P,$$
$$-\lambda_1^{(1)} \leq 0, \quad -\lambda_1^{(0)} \leq 0, \quad -\lambda_2^{(1)} \leq 0, \quad -\lambda_2^{(0)} \leq 0,$$
$$\lambda_2^{(1)} - \lambda_1^{(1)} \leq 0, \quad \lambda_2^{(0)} - \lambda_1^{(0)} \leq 0$$

*Proof of Theorem 2.* By writing out the analytical close-form of the KL divergence between two Gaussian distributions, the optimization can be written as:

$$\min_{\Sigma_0, \Sigma_1 \in \mathbb{S}} \mathrm{tr}((\Sigma_1 + vI)^{-1}(\Sigma_0 + uI)) + \mathrm{tr}((\Sigma_0 + uI)^{-1}(\Sigma_1 + vI)) +$$

$$(\bar{g}^{(1)} - \bar{g}^{(0)})^\top \left((\Sigma_1 + vI)^{-1} + (\Sigma_0 + uI)^{-1}\right) (\bar{g}^{(1)} - \bar{g}^{(0)})$$

subject to

$$\Sigma_0 \Sigma_1 = \Sigma_1 \Sigma_0,$$
$$p \cdot \mathrm{tr}(\Sigma_1) + (1 - p) \cdot \mathrm{tr}(\Sigma_0) \leq P,$$
$$\Sigma_1 \succeq \mathbf{0},$$
$$\Sigma_0 \succeq \mathbf{0}.$$
$$(13)$$

By the commutative constraint on the two positive semidefinite matrices $\Sigma_1$ and $\Sigma_0$, we know that we can factor these two matrices using the same set of eigenvectors. We thus write:

$$\Sigma_0 = Q^\top \mathrm{diag}(\lambda_1^{(0)}, \ldots, \lambda_d^{(0)})Q,$$
$$\Sigma_1 = Q^\top \mathrm{diag}(\lambda_1^{(1)}, \ldots, \lambda_d^{(1)})Q,$$
$$(14)$$

where $Q \in \mathbb{R}^{d \times d}$ is an orthogonal matrix and the eigenvalues $\lambda_i^{(0)}, \lambda_i^{(1)}$ are nonnegative and decreasing in value.

Using this alternative expression of $\Sigma_1$ and $\Sigma_0$, we can express the optimization in terms of $\{\lambda_i^{(1)}\}, \{\lambda_i^{(0)}\}, Q$:

$$\min_{\{\lambda_i^{(1)}\}, \{\lambda_i^{(0)}\}, Q} \sum_{i=1}^d \frac{\lambda_i^{(0)} + u}{\lambda_i^{(1)} + v} + \sum_{i=1}^d \frac{\lambda_i^{(1)} + v}{\lambda_i^{(0)} + u} +$$

$$\left[Q(\bar{g}^{(1)} - \bar{g}^{(0)})\right]^\top \mathrm{diag}\left(\ldots, \frac{1}{\lambda_i^{(0)} + u} + \frac{1}{\lambda_i^{(1)} + v}, \ldots\right) Q(\bar{g}^{(1)} - \bar{g}^{(0)})$$

$$\text{subject to} \quad p(\sum_{i=1}^{d} \lambda_i^{(1)}) + (1-p)(\sum_{i=1}^{d} \lambda_i^{(0)}) \leq P$$

$$-\lambda_i^{(1)} \leq 0, \ \forall i \in [d]$$

$$-\lambda_i^{(0)} \leq 0, \ \forall i \in [d].$$

$$\lambda_i^{(1)} \geq \lambda_j^{(1)}, \forall i < j.$$

$$\lambda_i^{(0)} \geq \lambda_j^{(0)}, \forall i < j.$$

$$Q \ \text{orthogonal}.$$

For any fixed feasible $\{\lambda_i^{(1)}\}, \{\lambda_i^{(0)}\}$, we see that the corresponding minimizing $Q$ will set its first row to be the unit vector in the direction of $\Delta g$. Thus by first minimizing $Q$, the optimization objective reduces to:

$$\sum_{i=1}^{d} \frac{\lambda_i^{(0)} + u}{\lambda_i^{(1)} + v} + \sum_{i=1}^{d} \frac{\lambda_i^{(1)} + v}{\lambda_i^{(0)} + u} + \frac{g}{\lambda_1^{(0)} + u} + \frac{g}{\lambda_1^{(1)} + v}$$

$$= \sum_{i=2}^{d} \frac{\lambda_i^{(0)} + u}{\lambda_i^{(1)} + v} + \sum_{i=2}^{d} \frac{\lambda_i^{(1)} + v}{\lambda_i^{(0)} + u} + \frac{\lambda_1^{(1)} + v + \|\Delta g\|_2^2}{\lambda_1^{(0)} + u} + \frac{\lambda_1^{(0)} + u + \|\Delta g\|_2^2}{\lambda_1^{(1)} + v}$$

We see that for the pair of variable $(\lambda_i^{(1)}, \lambda_i^{(0)})$ ($i \geq 2$), the function $\frac{\lambda_i^{(0)}+u}{\lambda_i^{(1)}+v} + \frac{\lambda_i^{(1)}+v}{\lambda_i^{(0)}+u}$ is strictly convex over the line segment $p\lambda_i^{(1)} + (1-p)\lambda_i^{(0)} = c$ for any nonnegative $c$ and attains the the minimum value at $\lambda_i^{(1)} = 0$ when $u < v$ and $\lambda_i^{(0)} = 0$ when $u \geq v$. Suppose without loss of generality $u \geq v$, then for the optimal solution we must have $\lambda_i^{(0)} = 0$ for all $i \geq 2$. Under this condition, we notice that the function $m(x) = \frac{u}{x+v} + \frac{x+v}{u}$ is strictly convex on the positive reals. Thus for all $\left\{\lambda_i^{(1)}\right\}$ that satisfies $\sum_{i=2}^{d} \lambda_i^{(1)} = c$ for a fixed nonnegative $c$, by Jensen inequality, we have

$$\frac{1}{d-1} \sum_{i=2}^{d} \left( \frac{u}{\lambda_i^{(1)} + v} + \frac{\lambda_i^{(1)} + v}{u} \right)$$

$$= \frac{1}{d-1} \sum_{i=2}^{d} m(\lambda_i^{(1)})$$

$$\geq m\left( \frac{1}{d-1} \sum_{i=2}^{d} \lambda_i^{(1)} \right)$$

$$= m\left( \frac{c}{d-1} \right).$$

From this, we see that the optimal solution's variables $\{\lambda_i^{(1)}\}$ must take on the same value ($\frac{c}{d-1}$) for all $i \geq 2$. The case when $u \leq v$ is similar. As a result, we have proved that at the optimal solution, we must have:

$$\lambda_i^{(0)} = \lambda_j^{(0)} \text{ and } \lambda_i^{(1)} = \lambda_j^{(1)}, \text{ for } i, j \geq 2.$$

Hence, the optimization problem over the $2d$ variables $\left\{\lambda_i^{(1)}\right\}_{i=1}^d \bigcup \left\{\lambda_i^{(0)}\right\}_{i=1}^d$ can be reduced to an optimization problem over the four variables $\left\{\lambda_1^{(1)}, \lambda_2^{(1)}, \lambda_1^{(0)}, \lambda_2^{(0)}\right\}$:

$$\min_{\lambda_1^{(0)}, \lambda_1^{(1)}, \lambda_2^{(0)}, \lambda_2^{(1)}} (d-1)\frac{\lambda_2^{(0)} + u}{\lambda_2^{(1)} + v} + (d-1)\frac{\lambda_2^{(1)} + v}{\lambda_2^{(0)} + u} + \frac{\lambda_1^{(0)} + u + \|\Delta g\|_2^2}{\lambda_1^{(1)} + v} + \frac{\lambda_1^{(1)} + v + \|\Delta g\|_2^2}{\lambda_1^{(0)} + u}$$

$$(15)$$

$$\text{subject to} \quad p\lambda_1^{(1)} + p(d-1)\lambda_2^{(1)} + (1-p)\lambda_1^{(0)} + (1-p)(d-1)(\lambda_2^{(0)}) \leq P$$
$$-\lambda_1^{(1)} \leq 0$$
$$-\lambda_1^{(0)} \leq 0$$
$$-\lambda_2^{(1)} \leq 0$$
$$-\lambda_2^{(0)} \leq 0$$
$$\lambda_2^{(1)} - \lambda_1^{(1)} \leq 0$$
$$\lambda_2^{(0)} - \lambda_1^{(0)} \leq 0.$$

Given the optimal solution to the above 4-variable problem $(\lambda_1^{(0)*}, \lambda_2^{(0)*}, \lambda_1^{(1)*}, \lambda_2^{(1)*})$, we can set $Q$ to be any orthogonal matrix whose first row is the vector $\frac{\Delta g}{\|\Delta g\|_2}$. Plugging this back into the expression of $\Sigma_1$ and $\Sigma_0$ in Equation (14) gives us the final result.

Thus the proof is complete. $\qquad\square$

*Remark* (Interpreting the optimal $\Sigma_1^*$ and $\Sigma_0^*$). From the form of the optimal solution in (12), we see that the optimal covariance matrices are both linear combinations of two terms: a rank one matrix $(\Delta g)(\Delta g)^\top$ and the identity matrix $I_d$. Because a zero-mean Gaussian random vector with convariance matrix $(A + B)$ can be constructed as the sum of two independent zero-mean Gaussian random vectors with covariance matrices $A$ and $B$ respectively, we see that the optimal additive noise random variables $\eta^{(1)}$ and $\eta^{(0)}$ each consist of two independent components: one random component lies along the line that connects the positive and negative gradient mean vectors (whose covariance matrix is proportional to $\Delta g \Delta g^\top$); the other component is sampled from an isotropic Gaussian. The use of the first random directional component and the fact that the isotropic Gaussian component have different variance scaling for the positive and negative class clearly distinguishes `Marvell` from the isotropic Gaussian baseline `iso`.

*Remark* (How to solve). By analyzing the KKT condition of this four variable problem we can find that the optimal solution must exactly lie on the hyperplane $p\lambda_1^{(1)} + p(d-1)\lambda_2^{(1)} + (1-p)\lambda_1^{(0)} + (1-p)(d-1)\lambda_2^{(0)} = P$. From the proof above we additionally know that $\lambda_2^{(1)*} = 0$ if $u < v$ and $\lambda_2^{(0)*} = 0$ if $u \geq v$. Thus the problem is further reduced to a 3-variable problem. If we consider keeping one of the 3 remaining variables' values fixed, then the feasible region becomes restricted to a line segment. We can simply perform a line search optimization of the convex objective and find the optimal values for the remaining two free variables. We can then alternate over which one of the three variables to fix and optimize over the other two. This optimization procedure would eventually converge and give us the optimal solution. This approach mimics the Sequential Minimal Optimization technique used to solve the dual of the support vector machine (SVM).

### A.5 MARVELL ALGORITHM DESCRIPTION

We use $\mathbf{1} \in \mathbb{R}^d$ to denote the vector with 1 in each coordinate. We use $g[j]$ to denote the $j$-th row of the matrix $g$ and $y[j]$ to denote the $j$-th coordinate of the vector $y$.

---

**Algorithm 1:** `Marvell` algorithm

**input** : $g \in \mathbb{R}^{B \times d}$,    a size-$B$ batch of unperturbed gradients
           $y \in \{0,1\}^B$,   the label for each example in the batch
           $s$,                privacy hyperparameter for the power constraint $P$

**output:** $\tilde{g} \in \mathbb{R}^{B \times d}$,    batch of perturbed gradients

`// Step 1: estimate the optimization constants from the batch`
`   gradients using maximum likelihood estimation (MLE)`

1   $p \leftarrow \frac{\mathbf{1}^T y}{d}$ ;                                            `/* positive fraction */`

2   $\bar{g}^{(1)} \leftarrow \frac{1}{\mathbf{1}^T y} g^T y$ ;                                 `/* positive mean */`

3   $\bar{g}^{(0)} \leftarrow \frac{1}{B - \mathbf{1}^T y} g^T (\mathbf{1} - y)$ ;                      `/* negative mean */`

4   $\Delta g = \bar{g}^{(1)} - \bar{g}^{(0)}$;

5   $v \leftarrow \frac{1}{d \cdot (\mathbf{1}^T y)} \sum_{j=1}^{B} y[j] \cdot \left\| g[j] - \bar{g}^{(1)} \right\|_2^2$ ;        `/* positive convariance */`

6   $u \leftarrow \frac{1}{d \cdot (B - \mathbf{1}^T y)} \sum_{j=1}^{B} (1 - y[j]) \cdot \left\| g[j] - \bar{g}^{(0)} \right\|_2^2$ ;    `/* negative convariance */`

7   $P \leftarrow s \cdot \|\Delta g\|_2^2$ ;                     `/* power constraint hyperparameter */`

`// Step 2: optimize the four-variable problem`

8   **if** $u < v$ **then**

9       $\lambda_2^{(1)} \leftarrow 0$ ;                     `/* this variable is optimal at 0 */`

10      Randomly initialize the optimization variables $\lambda_1^{(1)}, \lambda_1^{(0)}, \lambda_2^{(0)}$ in the feasible region in (15).

11   **else**

12      $\lambda_2^{(0)} \leftarrow 0$ ;                     `/* this variable is optimal at 0  */`

13      Randomly initialize the optimization variables $\lambda_1^{(1)}, \lambda_2^{(1)}, \lambda_1^{(0)}$ in the feasible region in (15).

14   **end**

15   **while** *not converged* **do**

16      Fix one of the newly updated optimization variables;

        `// The 4-variable optimal solution lies on a hyperplane in` $\mathbb{R}^4$ `(see`
           `Appendix A.4 Remark) so fixing two variables gives us a`
           `line-segment`

17      Update the remaining two optimization variables by performing 1-d line-search
      minimization of the convex function (15) while satisfying the constraints;

18   **end**

`// Step 3: compute the optimal covariance matrices`

19   $\Sigma_1^* \leftarrow \frac{\lambda_1^{(1)} - \lambda_2^{(1)}}{\|\Delta g\|_2^2} (\Delta g)(\Delta g)^\top + \lambda_2^{(1)} I_d$;

20   $\Sigma_0^* \leftarrow \frac{\lambda_1^{(0)} - \lambda_2^{(0)}}{\|\Delta g\|_2^2} (\Delta g)(\Delta g)^\top + \lambda_2^{(0)} I_d$;

`// Step 4: perturb the gradients`

21   $\tilde{g} \leftarrow \mathbf{0}_{B \times d}$ ;        `/* an empty matrix to store the perturbed gradients */`

22   **for** $j \leftarrow 1$ **to** $B$ **do**

23      **if** $y[j] = 1$ **then**

24         $\tilde{g}[j] \leftarrow g[j] + \eta^{(1)}$, where $\eta^{(1)} \sim \mathcal{N}(\mathbf{0}, \Sigma_1^*)$;

25      **else** `//` $y[j] = 0$

26         $\tilde{g}[j] \leftarrow g[j] + \eta^{(0)}$, where $\eta^{(0)} \sim \mathcal{N}(\mathbf{0}, \Sigma_0^*)$;

27      **end**

28   **end**

---

### A.5.1 MARVELL TIME COMPLEXITY

The `Marvell` algorithm in 1 consists of the following steps:

Step 1. Compute the positive and negative gradient mean and their difference (line 1 - 4). This amounts to averaging over at most $B$ numbers over each of the d gradient dimensions. Thus would have a time complexity of $O(Bd)$.

Step 2. Compute the positive and negative covariance constants $u$ and $v$ (line 5, 6). This operation also takes $O(Bd)$ as it averages over squares of coordinate differences.

Step 3. Use the results from step 1 and 2, solve the 4-variable optimization problem (line 7-18). Here because the constant size of the number of optimization variables. Solving this problem up to a fixed precision takes constant time $O(1)$.

Step 4. Perform the actual random perturbation (line 19 - 28). Because of the additive structure of each class's covariance matrix, for every example's gradient in the batch, we can independently sample one random Gaussian vector with rank-1 covariance and also another spherical Gaussian random vector and add both vectors to this gradient for perturbation. This step would take $O(d)$ for each example and thus takes $O(Bd)$ in total.

As a result, the entire Marvell algorithm has a time complexity of $O(Bd)$. This can be further sped up through parallel computation using multi-threading/multi-core. Considering backpropagation through the cut layer would also require $O(Bd)$ time complexity, the Marvell algorithm would not slow down the split-learning process in any significant way at all.

### A.5.2 MARVELL EMPIRICAL RUN TIME

Empirically, in Table 1, we present the average amount of time it takes to run Marvell with the privacy hyperparameter value $s = 4$ for the three models (one for each dataset) we considered in our experiments ($s$ value is chosen as it achieves good privacy protection as shown with purple line in Figure 3, 5, 6). We also compare it to the average time it takes to run one update iteration. We additionally include the 95% confidence interval for the mean estimator of the run time. Here we notice that the total run time of Marvell only takes up a very small amount of the total training time for each method, further corroborating our algorithm's time efficiency advantage.

|  | batch size $B$ | cut layer feature dimension $d$ | average `Marvell` run time (seconds / run) | average update time (seconds / iteration) |
|---|---|---|---|---|
| ISIC | 128 | $1600$ $= 5 \times 5 \times 64$ | $1.79 \times 10^{-2}$ $\pm\ 7.03 \times 10^{-5}$ | $3.94 \times 10^{-1}$ $\pm\ 4.05 \times 10^{-3}$ |
| Criteo | 1024 | 128 | $1.68 \times 10^{-2}$ $\pm\ 8.67 \times 10^{-5}$ | $4.84$ $\pm\ 3.28 \times 10^{-1}$ |
| Avazu | 32768 | 128 | $1.75 \times 10^{-2}$ $\pm\ 1.47 \times 10^{-4}$ | $6.22$ $\pm\ 3.71 \times 10^{-1}$ |

Table 1: Average run time of `Marvell` ($s = 4$) and average total update time per iteration for models trained on ISIC, Criteo, and Avazu.

## A.6 Data Setup and Experimental Details

We first describe how we preprocess each of the datasets in A.6.1. We then describe the model architecture used for each dataset in A.6.2. Finally, we describe what the training hyperparameters are used for each dataset/model combination and the amount of compute used for the experiments in A.6.3.

### A.6.1 Dataset preprocessing

**[Criteo]** Every record of Criteo has 27 categorical input features and 14 real-valued input features. We first replace all the NA values in categorical features with a single new category (which we represent using the empty string) and replace all the NA values in real-valued features with 0. For each categorical feature, we convert each of its possible value uniquely to an integer between 0 (inclusive) and the total number of unique categories (exclusive). For each real-valued feature, we linearly normalize it into $[0, 1]$. We then randomly sample 10% of the entire Criteo publicly provided training set as our entire dataset (for faster training to generate privacy-utility trade-off comparision) and further make the subsampled dataset into a 90%-10% train-test split.

**[Avazu]** Unlike Criteo, each record in Avazu only has categorical input features. We similarly replace all NA value with a single new category (the empty string), and for each categorical feature, we convert each of its possible value uniquely to an integer between 0 (inclusive) and the total number of unique categories (exclusive). We use all the records in provided in Avazu and randomly split it into 90% for training and 10% for test.

**[ISIC]** The official SIIM-ISIC Melanoma Classification dataset has a total 33126 of skin lesion images with less than 2% positive examples. Because for image classification model training it is desirable to use a batch size of $\sim 10^2$, it is highly likely that there won't be any positive examples sampled in a batch of such size. Thus to make the label leakage problem more severe, we modify the dataset by retaining all the 584 positive examples and randomly choosing $584 \times 9$ examples out of all the negative examples. By doing this, we enforce that there are 10% positive examples in this modified dataset. We randomly split these 5840 examples into a 80%-20% training and test split. We also resize the images to size $84 \times 84$ for efficient model training.

### A.6.2 Model architecture details

**[Criteo, Avazu]** We use a popular deep learning model architecture WDL (Cheng et al., 2016) for online advertising. Here the deep and wide part each first processes the categorical features in a given record by applying an embedding lookup for every categorical feature's value. We use an embedding dimension of 4 for the deep part and embedding dimension of 1 for the wide part. After the lookup, the deep/wide embeddings are then concatenated with the continuous features to form the raw input vectors for both the deep part and wide part respectively. (This step is skipped for Avazu as it has no continuous features.) Then the wide part computes the wide part logit value through a real-valued linear function (with bias) of its raw input vectors, while the deep part processes its raw input features using 6 ReLU-activated 128-unit MLP layers before producing a single deep part logit. The two logits are summed up to form the final logic value. The cut layer is after the output of the 3rd ReLU layer on the deep part.

**[ISIC]** Every input image after resizing is of size $84 \times 84 \times 3$. We use a convolutional model with 6 convolutional layers each with 64 channels $3 \times 3$ filter size with $1 \times 1$ stride size. Each convolutional layer is followed by a ReLU activation function whose output is then max pooled with $2 \times 2$ window and stride size $2 \times 2$. The max-pooled output of the 6th layer is then flattened and pass into a 64-unit ReLU-activated MLP layer before finally being linearly transformed into a single logit score. The cut layer is after the output of the 4th max pool layer. Thus the cut layer feature and gradient are both of shape $5 \times 5 \times 64$.

### A.6.3 Model training details

Because the protection mechanism requires adding noise to the cut layer gradient, the induced variance of the gradients of non-label party's $f$-parameters becomes larger. Thus to ensure smooth

optimization and sufficient training loss minimization, we use a slightly smaller learning rate than what is normally used.

**[Criteo]** We use the Adam optimizer with a batch size of 1024 and a learning rate of $1e{-}4$ throughout the entire training of 5 epochs (approximately 20k stochastic gradient updates).

**[ISIC]** We use the Adam optimizer with a batch size of 128 and a learning rate of $1e{-}5$ throughout the entire training of 1000 epochs (approximately 35k stochastic gradient updates).

**[Avazu]** We use the Adam optimizer with a batch size of 32768 and a learning rate of $1e{-}4$ throughout the entire training of 5 epochs (approximately 5.5k stochastic gradient updates).

We conduct our experiments over 16 Nvidia 1080Ti GPU card. Each run of Avazu takes about 11 hours to finish on a single GPU card occupying 8GB of GPU RAM. Each run of Criteo takes about 37 hours to finish on a single GPU card using 5 GB of GPU RAM. Each run of ISIC takes about 12 hours to finish on a single GPU card occupying 4GB of GPU RAM.

### A.7 COMPLETE EXPERIMENTAL RESULTS

#### A.7.1 LEAK AUC PROGRESSION FOR AVAZU AND CRITEO

In addition to the leak AUC progression on ISIC shown in Figure 3 in the main paper, we also show the leak AUC progression on the Avazu and Criteo datasets throughout training here in Figure 5 and Figure 6. We similarly compare `Marvell` with different levels of protection strength ($s$ values) against the no protection baseline `no_noise`. As we can see, `Marvell` still achieves strong and flexible privacy protection on these two datasets against our label attacks at different model layers.

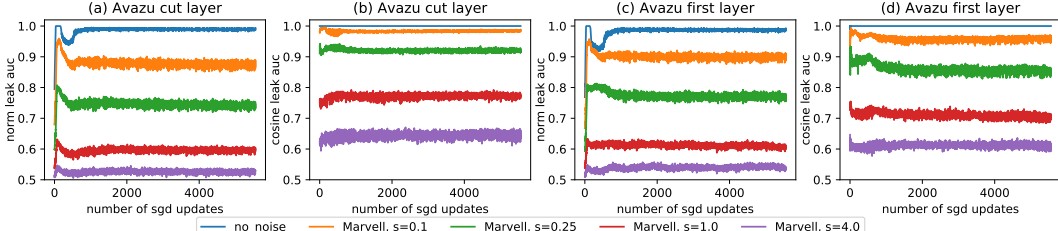

Figure 5: Norm and cosine leak AUC (computed every batch) at the cut layer and at the first layer of `no_noise` (no protection) vs. `Marvell` with different scale hyperparameter $s$ throughout the Avazu training.

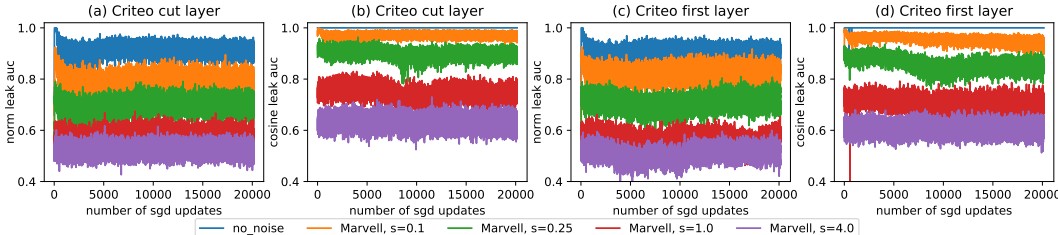

Figure 6: Norm and cosine leak AUC (computed every batch) at the cut layer and at the first layer of `no_noise` (no protection) vs. `Marvell` with different scale hyperparameter $s$ throughout the Criteo training.

#### A.7.2 COMPLETE PRIVACY-UTILITY TRADEOFFS

We show additional Privacy-Utility tradeoff results for all three datasets considered in this paper. (Some of the plots have already been shown in the main paper but we still include them here for completeness and ease of reference.) For each dataset, we compare the privacy-utility tradeoff over multiple measures of privacy and utility:

**Privacy**

We consider our introduced privacy metrics using the activation gradient from the cut layer and the first layer of the non-label party :

- 95% norm leak AUC at cut layer
- 95% cosine leak AUC at cut layer
- 95% norm leak AUC at first layer
- 95% cosine leak AUC at first layer

**Utility**

We consider three metrics of utility:

- training loss (train loss): the lowest loss achieved on the training set throughout training. This directly measures how much the random protection perturbation influences the optimization.
- test loss. Because we only control the training optimization stochastic gradient's variance, measuring test loss directly tells us how much impact the training optimization random perturbation influences beyond optimization but on the learned model's generalization ability.
- test AUC. As we are dealing with binary classification problem (where performance is commonly measured through test AUC), we also naturally consider it as a utility metric.

As shown in Figure 7,8,9, Marvell consistently outperforms the isotropic Gaussian baseline over all the different privacy-utility definitions. In addition, our proposed heuristic max_norm is also particularly effective against our identified norm and direction-based attacks.

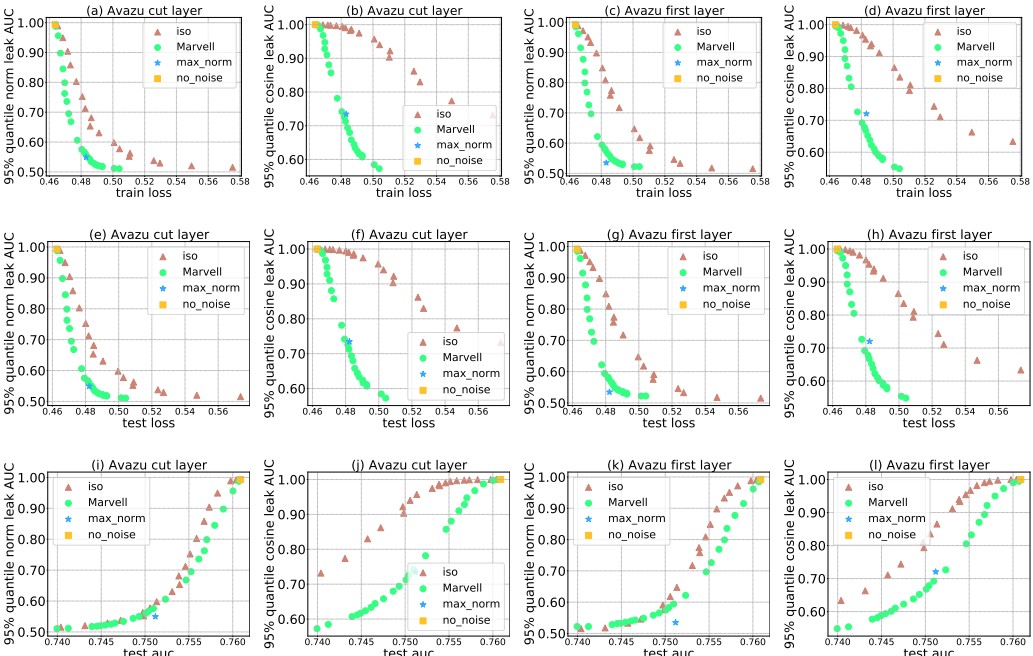

Figure 7: Privacy (norm and cosine leak AUC) vs Utility (train loss, test loss, and test AUC) trade-off of protection methods (Marvell, iso, no_noise, max_norm) at the cut layer and first layer on Avazu.

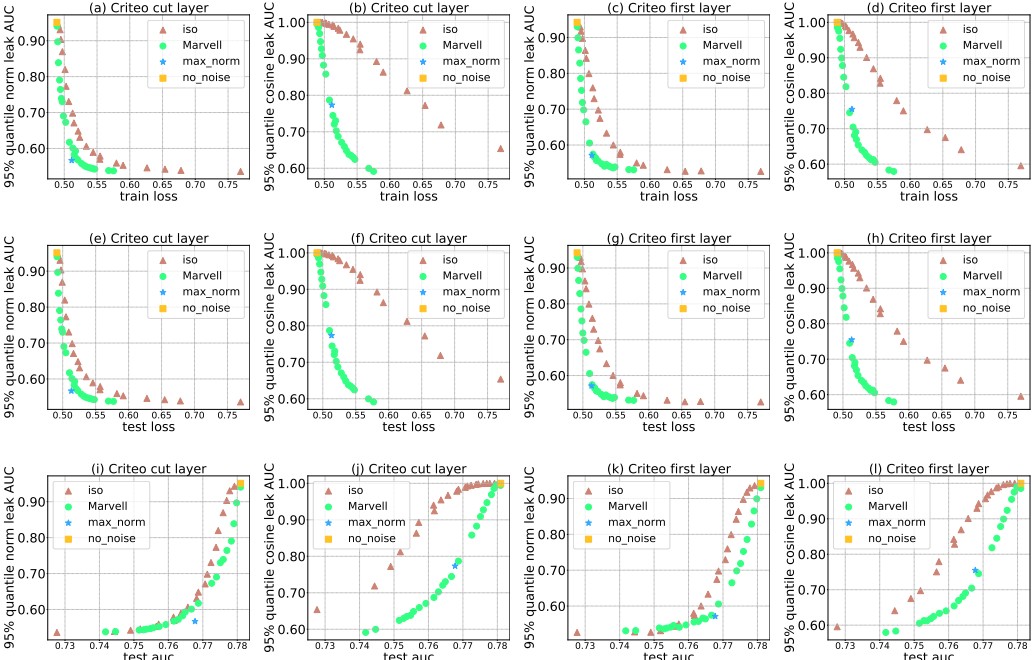

Figure 8: Privacy (norm and cosine leak AUC) vs Utility (train loss, test loss, and test AUC) trade-off of protection methods (Marvell, iso, no_noise, max_norm) at the cut layer and first layer on Criteo.

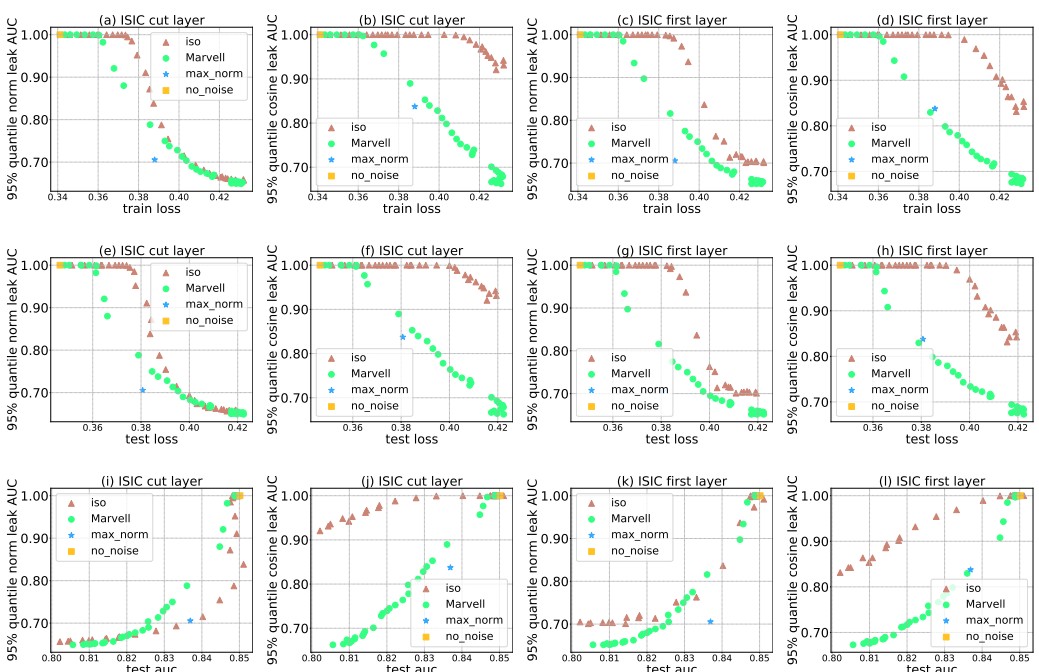

Figure 9: Privacy (norm and cosine leak AUC) vs Utility (train loss, test loss, and test AUC) trade-off of protection methods (`Marvell`, `iso`, `no_noise`, `max_norm`) at the cut layer and first layer on ISIC.

### A.7.3 IMPACT OF ACTIVATION GRADIENTS' LAYER LOCATION ON THE CORRESPONDING LEAK AUC

In Section 5.1, we have shown that the activation gradients of not only the cut layer but also the **first layer** can leak the label if no protection is applied. In this section, we analyze the effect of this layer location on the degree of label leakage under our proposed attacks measured through leak AUC. Here, for the convolutional neural network trained on ISIC, we have allocated 4 Conv-ReLU-MaxPool layers on the non-label party side. Here each of these 4 layers' activation (from ReLU output) gradients can be used to infer the label. In Figure 10, we plot the progression of norm and cosine leak AUC for layer 2 and 3 in addition to the first layer (layer 1) and the cut layer (layer 4) when applying no protection or Marvell with $s = 4$. We notice that when **no protection is applied, the norm and cosine leak AUC are approximately the same across the layers and both at a very high level, indicating a strong degree of label leakage**. In contrast, **when Marvell is applied, the leak AUCs are much lowered to a reasonable level of around** $0.6$ **throughout all the layers**. Empirically, we notice that the leak AUCs for a given training iteration are close among all the layers with a maximum difference of around $0.05$. Here it is *unclear whether the earlier layers' gradients would leak the labels more than the cut layer or not* — for norm leak AUC, the earlier layers have a higher AUC values (more leakage) than the cut layer, while for cosine leak AUC, the earlier layers have a lower AUC values (less leakage) than the cut layer. Further understanding the relationship between these leak AUC values across different layers is an open question motivated by our observations here.

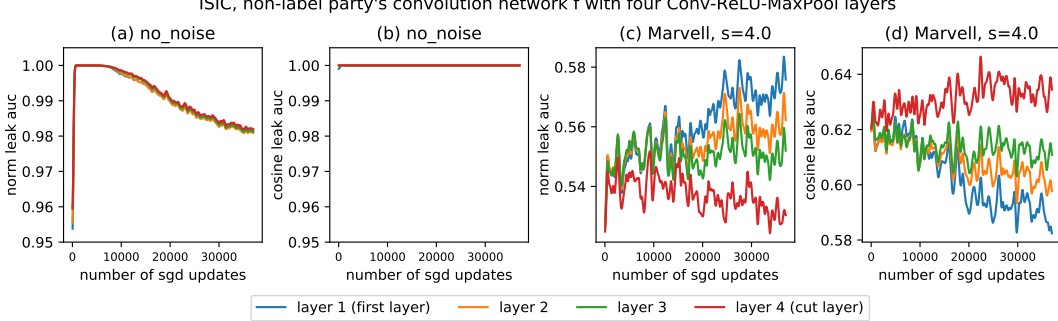

Figure 10: For a non-label party's four-layer convolutional architecture trained on ISIC, we plot the progression of the norm or cosine leak AUC computed using the activation gradients from each of the four layers throughout training in (a), (b) when no random protection is applied and in (c), (d) when using Marvell with privacy hyperparameter $s = 4$ at the cut layer (layer 4). The four curves overlap in (b) at the value of $1.0$. For each layer's data in each figure, a 1-d Gaussian kernel with standard deviation of 5 is convolved with the raw 1-d array of leak AUC values to smooth out the fluctuations and make the different layers' degree of leakage more visually distinct.

Beyond the four layer convolutional neural network trained on ISIC, we also train a new MLP model architecture on Criteo. Here instead of having 3 128-unit ReLU-activated MLP layers for the non-label party's function $f$ as described in Section A.6.2, we use 8 such layers for $f$ while keeping every other architectural component the same. We plot the progression of leak AUC values on each of the eight layers in Figure 11. Here the observations are very similar to what we see in Figure 10, confirming our aforementioned conclusions in this section.

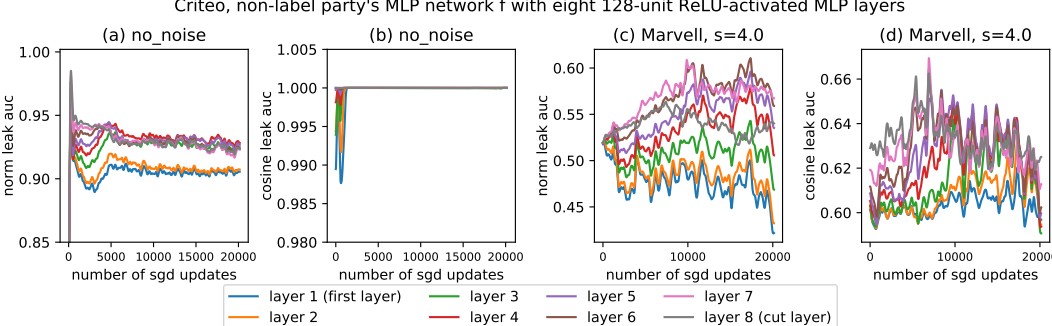

Figure 11: For a non-label party's eight-layer MLP architecture trained on Criteo, we plot the progression of the norm or cosine leak AUC computed using the activation gradients from each of the eight layers throughout training in (a), (b) when no random protection is applied and in (c), (d) when using `Marvell` with privacy hyperparameter $s = 4$ at the cut layer (layer 4). The eight curves overlap in (b) at the value of $1.0$. For each layer's data in each figure, a 1-d Gaussian kernel with standard deviation of 100 is convolved with the raw 1-d array of leak AUC values to smooth out the fluctuations and make the different layers' degree of leakage more visually distinct.

## A.8 Reasons why we don't use local DP in our setup

It is possible to use a local differential privacy definition to analyze our problem setup if we treat any single training example as a dataset and two examples (dataset) are adjacent if they share the same $X$ but have the opposite label $y$. Then one can analyze the activation gradient computation function's local DP properties. However, we choose **not to use local DP in our setup** for the following two reasons:

1. We are not aware of any attacks associated with local DP that are suitable for practical use in our problem setup. In contrast, our leak AUC privacy quantification metrics have corresponding concrete, realistic attack methods that the non-label party can use in practice.
2. Even if there exists a practical attack associated with local DP, this attack would require **example-level side information** as the activation gradient distributions would be example-specific. Distinguishing between such gradient distributions for a pair of adjacent examples (with same $X$ but opposite label $y$) would require knowledge about the two distributions specific to this example pair itself. As we have mentioned in Section 3.2, we **do not assume the non-label party would realistically have access to such fine-grained knowledge**. Instead, in our setup, the non-label party uses a scoring function that only takes in the communicated cut layer gradient without any additional information.

