# OpenReview forum: "Label Leakage and Protection in Two-party Split Learning"
_ICLR.cc/2022/Conference — ICLR 2022 Poster_

### Official Review · Reviewer_SgGa · 2021-10-30

**Correctness:** 3
**Technical Novelty And Significance:** 3
**Empirical Novelty And Significance:** 4
**Recommendation:** 6
**Confidence:** 2

**Main Review:**

## Strengths

- Label privacy is an overlooked problem that has many practical applications. This paper provides a formulation for this problem together with a fairly strong attacks and a promising mitigation.

## Weaknesses

- I think the current split learning setting seems strange: if our goal is strictly to protect just the labels and not the feature vectors, it would also be fine to the non-label party to send the features directly to the label party; then, the latter can train the entire model by itself without leaking any label at all. This seems much better for label-privacy and also for communication complexity.

## Comments for Authors

- As stated above, it would be good to justify the current split learning setting more. For example, is the goal here to also protect the X? If so, then shouldn't we also employ some method for that? And how would such a method, in conjunction with those presented in the paper, effect the utility?

- It is stated (on pages 2 and 3) that "DP and its variants are not directly applicable metric in our setting". I do not think this statement is true. In fact, *local* DP would still be applicable to non-aggregate data. The paper [Ghazi et al. 2021] you cited also yield label DP in the local setting without aggregation (by flipping the labels). Similarly, the isotropic Gaussian baseline discussed in the paper also achieve some level of local DP. I think this point has to be clarified in the revised version.


**Summary Of The Paper:**

## Summary of Contributions

This paper studies the multiparty setting where there are two parties: the first party (aka non-label party) holds the feature vectors $X_i$ of user $i$ whereas the second party (aka label party) holds the corresponding labels $y_i \in \{0, 1\}$. Together they wish to jointly train a model in such a way that the non-label party does not learn of the users' labels. This is especially relevant in online advertisement (where the label party corresponds to the advertiser who knows whether each user converts and the non-label party is e.g. the publisher who shows the ads) and in medical applications (where the label party is the hospital).

The paper focuses on *split training* which is a setting where the model is divided among two parties with the first layers (denoted as a function $f$) is with the first party and the remaining layers (denoted as a function $h$) are with the second party. Here the training is performed as follows. First, the non-label party computes $f(X)$ (referred to as the "cut layer") and send this to the label party. The label party then compute the gradient w.r.t. $h$ and update its parameters and furthermore it computes the gradient $g$ w.r.t. the cut layer $f(X)$ and sends it back to the non-label party. The non-label party can then use backpropagation starting with $g$ to update its own parameters. The main concern tackled in this paper is that, in such a scheme, the gradient $g$ sent back to the non-label party can leak the information about the label. Specifically, the authors observe and experimentally show that the following two attacks are very effective in predicting the labels given the intermediate gradient $g$:
- Consider the norm $\\|g\\|_2$ and make prediction based on whether it exceeds a certain threshold.
- Consider the cosine similarity between $g$ and another (fixed) gradient, and then threshold.
In fact, the two above attacks get near-perfect predictions in some examples. The exact measure they use to evaluate their model is the area-under-curve (AUC) of the (false positive rate, true positive rate) curve.

The authors then propose to add noise in order to mitigate such attacks. The baseline here would be to add isotopic Gaussian noise (with a certain scale). The authors then formulate an optimization problem aiming to achieve better privacy-utility tradeoff. Roughly speaking, this is an optimization problem of the form "minimize AUC" such that "utility is at least ...". It is hard to make this precise (and it is probably inefficient), so the authors use certain proxy for both the objective and the constraint. For the objective, the authors show that AUC is upper bound by a certain quantity involving the KL divergence of the distributions of the gradients from the two labels. For the utility constraint, the authors instead try to restrict the amount of the noise added by having an upper bound on the trace of their covariance matrices. Furthermore, the authors show that when the (unperturbed) gradient distributions are assumed to be Gaussians and only Gaussian noises are considered, then the optimization problem simplifies greatly to just a single constant-size optimization problem (Theorem 2). This final form constitutes their so-called Marvell algorithm. Finally, the paper concludes by several experimental results showing that Marvell is very effective against protecting the two aforementioned attacks in real-world datasets and models, while preserving reasonable level of privacy.

**Summary Of The Review:**

## Recommendation

I think formalizing attack vectors and proposing mitigations for label privacy is an important contribution that may have a large impact. On the other hand, the current formulation/attack/mitigation is tailored towards the specific split learning setting which (as stated above) might require more motivation. Taken both into account, I'm giving a weak accept for now.

---

> ### Author Response · Authors · 2021-11-17
> **Response to Reviewer SgGa Part 2/2**
>
> **[Why do we state DP and Label DP cannot be directly used?]** Ghazi et al (2021) use Randomized Response with Prior (RRWithPrior) as a subroutine for their main training algorithm Multi-Stage Training (LP-MST). The authors prove in Theorem 4 that with RRwithPrior satisfying $\epsilon$-DP, the LP-MST algorithm would satisfy $\epsilon$-label DP. However, we want to point out that this label DP property of their machine learning algorithm LP-MST is **not in a local setting**, because their definition of adjacent datasets in Definition 2.2 is between two **global** multi-example datasets which differ only over an example’s label. In fact the original paper only mentions local DP when describing the Randomized Response mechanism but not the algorithm LP-MST. In our discussion, we use the term **aggregate functions** to describe mechanisms such as LP-MST. Here by aggregate functions we mean these mechanisms operate on the dataset level by taking in an entire dataset of examples as input and returning a model. In contrast, the cut-layer activation gradient computation mechanism we care about in our setup is not an aggregate function since it operates **on a single example level** and returns the example’s corresponding cut-layer gradient as output. Since in the canonical machine learning context definitions of DP and Label DP both focus on aggregate functions, we believe they are not appropriate to be used without significant modifications to our problem setting.
>
> **[Why don't we use Local DP?]** It is indeed possible to use a local differential privacy definition to analyze our problem setup if we treat any single training example as a dataset and two examples (dataset) are adjacent if they share the same $X$ but have the opposite label $y$. Then one can analyze the activation gradient computation function’s local DP properties. However, we choose **not to use local DP in our setup** for two reasons (we have also included this discussion in the updated Appendix A.8):
>
> 1. We are not aware of any attacks associated with local DP that are suitable for practical use in our problem setup. In contrast, our leak AUC privacy quantification metrics have corresponding concrete, realistic attack methods that the non-label party can use in practice.
>
> 2. Even if there exists a practical attack associated with local DP, this attack would require **example-level side information** as the activation gradient distributions would be example-specific. Distinguishing between such gradient distributions for a pair of adjacent examples (with same $X$ but opposite label $y$) would require knowledge about the two distributions specific to this example pair itself. As we have mentioned in Section 3.2, we **do not assume the non-label party would realistically have access to such fine-grained knowledge**. Instead, in our setup, the non-label party uses a scoring function that only takes in the communicated cut layer gradient without any additional information.

---

> > ### Comment · Reviewer_SgGa · 2021-11-19
> > **Re Response to Reviewer SgGa**
> >
> > I would like to thank the authors for the response(s). They answer most of my questions/comments. A few additional minor comments:
> > - I agree that evaluating a method for protecting labels alone first is reasonable (for a first paper in this topic), and perhaps adding methods to protect $X$ can be considered as a future work.
> > - Regarding Ghazi et al.'s: I'm fairly sure that their RRWithPrior satisfies local DP. You're right that they didn't say this in the paper but since it is just a variant of the vanilla Randomized Response (which definitely satisfies local DP), it should also satisfy local DP.
> > - Regarding Local DP in general: I think the second point you gave is quite interesting and should be highlighted more. Another quick comment is that I was thinking the other way around: local DP should give the upper bound on KL divergence between $\tilde{P}^{(0)}, \tilde{P}^{(1)}$ and then applying Theorem 1, this should imply that any $\epsilon$-local DP algorithm immediately gives an upper bound on AUC.

---

> > > ### Author Response · Authors · 2021-11-23
> > > **Followup Response to Reviewer SgGa**
> > >
> > > Thank you for following up!
> > > - We agree that RRWithPrior also satisfies local DP; by noting that Randomized Response satisfies local DP in our response (Part 2/2) we meant that both RR and RRWithPrior satisfy it.
> > > - Thanks for suggesting using the local DP property to upper bound the KL divergence between $P^{(0)}$ and $P^{(1)}$ used in our paper. We want to note that $P^{(1)}$ and $P^{(0)}$ in the local DP definition would be specific to **a given raw feature** $X$’s perturbed gradient distribution when its label $y$ is $1$ or $0$. However, the $P^{(1)}$ and $P^{(0)}$ in our setup refer to the **population distributions** of the perturbed positive and negative gradient. Reasoning about how to go from the relationship between example-level perturbed distributions (in local DP) to the relationship between population-level perturbed distributions (considered in our setup) would be an interesting direction of future work.
> > > - Thanks also for suggesting to highlight our choice of not using local DP for privacy quantification. We have added another sentence in Section 3.2’s paragraph on **Side Information**, where we have additionally referred the reader to Appendix A.8 for our newly added extended explanation around local DP.

---

> > > > ### Comment · Reviewer_SgGa · 2021-11-24
> > > > **Re Followup Response to Reviewer SgGa**
> > > >
> > > > Thanks the authors for the reply. Regarding the second point, doesn't the convexity of KL divergence means that a bound on example-level KL divergence (between $P^{(0)}$ and $P^{(1)}$) implies a bound on population-level KL divergence as well?

---

> > > > > ### Author Response · Authors · 2021-11-24
> > > > > **2nd Followup Response to Reviewer SgGa**
> > > > >
> > > > > Thanks for asking this question. Although we believe that understanding the details of how to go from local DP to our definition is beyond the scope of our current work, we also agree that this is an interesting direction so we provide our viewpoint below.
> > > > >
> > > > > We don’t believe that the convexity of KL divergence can be directly used to go from local DP to population-level KL divergence. One way for the convexity of KL to be used is if the perturbed population distribution is a simple mixture of the individual perturbed distribution. Here we want to emphasize that our definition of the population distribution of the perturbed positive (and negative) gradients **is not a simple mixture** of the individual perturbed positive (and negative) gradient distribution over every possible input feature $X$ as considered in local DP. This is because, in our definition of the population distribution, if a feature $X$ only appears with a positive label, we would only consider its individual perturbed gradient distribution under the positive label and mix it only in the population distribution of the perturbed **positive gradient but not the negative**. In contrast, in the setting of local DP, this feature $X$ would still have a hypothetical negative label and a corresponding individual perturbed negative gradient distribution. Thus this hypothetical individual negative distribution would still be used to construct the negative distribution mixture, yet it would never be used in our definition of the population distribution of the perturbed negative gradient.
> > > > >
> > > > > To illustrate this difference through notations, we denote the training dataset by $S = \\{(X_i, y_i) \\}\_{i=1}^{N}$ with every label $y_i \in \\{0, 1\\}$. The **population** distribution of perturbed **positive** gradients (under our definition) would be perturbing the examples $S_{+} := \\{(X, y) \in S \mid y = 1\\}$. On the other hand, in the local DP case, the mixture distribution of all possible individual perturbed positive gradient distributions would be perturbing over the example set $\\{(X, 1) \mid (X, y) \in S\\}$, which is likely a strict superset of the set $S_{+}$ considered in our population definition. Please let us know if you have any further questions.

---

> > > > > > ### Comment · Reviewer_SgGa · 2021-11-28
> > > > > > **Re 2nd Followup Response to Reviewer SgGa**
> > > > > >
> > > > > > I'd like to thank the authors again for the clear reply; I now understand why it is not easy to go from LDP bound on KL to the population bound as defined in the paper, and I agree that this is an interesting future direction.

---

> ### Author Response · Authors · 2021-11-17
> **Response to Reviewer SgGa Part 1/2**
>
> We thank the reviewer for their time in reviewing our submission. We respond to the points raised below:
>
> **[Why do we need split learning in our current setup?]** As we mentioned in the second paragraph of Section 1, one of the primary reasons for split learning is to not share the raw data between the two parties, which is why we don’t consider sending the features directly. Beyond this reason, split learning also enables the two entities to each individually choose their own desired model architecture for their prediction function ($f$ and $h$ respectively) (see also Vepakomma et al, 2018) as long as the cut layer feature dimension matches. In addition, split learning distributes the training and inference workload between the participating parties instead of burdening one party to perform all the work, which might also be preferable in certain real world scenarios.
>
> **[Should we also consider protecting the raw feature $X$?]** It is indeed correct that the raw feature $X$ on the non-label party could potentially also be leaked by communicating the transformed feature $f(X)$. As we discussed in the Section 2, a prior workshop paper by Vepakomma et al (2019) has proposed a preliminary approach to protect such leakage. They measure the privacy leakage through a distance correlation metric between $X$ and $f(X)$; however, this metric is not backed by a realistic attack method that can be used by the label party to actually recover the raw feature $X$. In our work, we focus on the alternate direction of protecting the raw label $y$ instead of $X$ which to the best of our knowledge has not been studied before. In fact, in certain application scenarios (as discussed in Section 1.1 second paragraph of (Ghazi et al 2021), the privacy of labels can be particularly sensitive and possibly more important than the privacy of raw input features. We believe studying the label leakage and protection in isolation from the feature leakage problem is an important initial step as understanding this problem alone is already a novel and underexplored direction. We agree that it would be an interesting direction of future work to study the interaction between any feature protection techniques (for $X$) and our proposed label protection technique Marvell (for $y$) and investigate their compounded effect on utility. In this case, we might need to measure the privacy alone using a two-tuple, with one number capturing the amount of feature leakage, and the other number capturing the amount of label leakage (for example using our proposed metric leak AUC).

---

### Official Review · Reviewer_GVaD · 2021-11-01

**Correctness:** 4
**Technical Novelty And Significance:** 3
**Empirical Novelty And Significance:** 3
**Recommendation:** 6
**Confidence:** 3

**Main Review:**

The paper considers a novel setting in split learning: stealing binary labels during training and the setting could not be solved directly by differential privacy. To begin with, the authors first explore the possibility of stealing label information via some heuristic observations of the gradient information of positive/negative examples. The effectiveness of the proposed attack is shown in the experiments. The authors then propose an optimized random perturbation approach to mitigate the attack and the defense is shown to achieve the best trade-offs among the baselines. The paper is well-written and clearly organized. Overall, I think the paper makes solid contributions.

My biggest concern is that the paper is limited to (class-unbalanced) binary classification. I carefully check the proposed attacks, but find it difficult to extend the multi-class classification. For example, observations 1-4 are not necessarily extended to multi-class classification.

Besides, since the proposed defense MARVELL at each update step requires solving an optimization problem, the (time) efficiency of MARVELL should also be discussed and demonstrated in the experiments. I am also curious about how the locations of the cut layer could affect the performance of the proposed attacks and defense (not limited to the first layer).


**Summary Of The Paper:**

The authors in the paper consider stealing the private label information from the party that does not know the label of training data during split training for binary classification. Specifically, the attack methods are based on differences of gradient information between positive and negative examples and the defense method MARVELL is based on optimal random perturbation solved by an optimization problem. Experimental results suggest that the proposed method could effectively defend against the label stealing attacks.

**Summary Of The Review:**

1. The paper makes solid contributions in privacy-preserving machine learning.
2. Time efficiency of the proposed methods should be discussed and more ablation study can be done.

---

> ### Author Response · Authors · 2021-11-17
> **Response to Reviewer GVaD Part 2/2**
>
> **[Layer location’s impact on label leakage]** We understand your question about “*the locations of the cut layer*” to be asking about how the label leakage metrics would change as we change the location of the layer (beyond the first and cut layer) whose gradients we input to the norm/direction-based scoring functions. (Please let us know if you are instead asking about actually changing the cut-layer location which would result in changing the predetermined split learning model architecture.) We have included a new section in Appendix A.7.3 to answer this question. We summarize our results here for completeness (please refer to the section for figures and more details): when no protection is applied, the norm and cosine leak AUCs are approximately the same across all the layers before the cut layer and are all at a very high level (very close to $1$). In contrast, when $\texttt{Marvell}$ is applied (with $s=4$), the leak AUCs across all the layers become consistently much lower to a reasonable level of around $0.6$ (with a maximum difference around $0.05$) for all the layers.

---

> > ### Comment · Reviewer_GVaD · 2021-11-19
> > **Response after rebuttal**
> >
> > Thanks for your response and it addresses my concerns.

---

> ### Author Response · Authors · 2021-11-17
> **Response to Reviewer GVaD Part 1/2**
>
> We thank the reviewer for their time in reviewing our submission. We respond to the points raised below:
>
> **[Summary correction]** Just to double-check, we believe you mean to say the setting we study **cannot** be solved directly using differential privacy.
>
> **[Limited to binary classification]** It is correct that our paper targets label privacy issues for binary classification split learning problems. As there are many real world problems (e.g., online advertising, disease prediction, recidivism prediction, loan prediction) that can be formulated as a binary classification problem, we believe this is a reasonable first step and don’t believe that this setting is too narrow. We have also identified multi-class label privacy as an open question in Section 6, and we agree that it would be an interesting direction of future study. As mentioned in the response to all reviewers (and discussed in more detail to Reviewer ercK), we note that we do not always require *imbalanced* binary classification problems either in terms of our attacks or protection methods.
>
> **[Time complexity of Marvell]** We provide a detailed time complexity analysis for Marvell here and have also included this discussion in the updated Appendix A.5.1. The Marvell algorithm, as presented in Appendix A.5, takes as input **1)** a batch of gradients $g\in \mathbb{R}^{B \times d}$, **2)** the label for each example $y \in \\{0,1\\}^B$, and **3)** the privacy hyperparameter $s$; and returns the batch of perturbed gradients $\widetilde{g} \in \mathbb{R}^{B \times d}$. Its computation consists of the following steps:
>
> 1. Compute the positive and negative gradient mean and their difference (line 1 - 4). This amounts to averaging over at most $B$ numbers over each of the d gradient dimensions. Thus it would have a time complexity of $O(Bd)$.
>
> 2. Compute the positive and negative covariance constants $u$ and $v$ (line 5, 6). This operation also takes $O(Bd)$ as it averages over squares of coordinate differences.
>
> 3. Using the results from step 1 and 2, solve the 4-variable optimization problem (line 7-18). Due to the constant size of the number of optimization variables, solving this problem up to a fixed precision takes constant time $O(1)$.
>
> 4. Perform the actual random perturbation (line 19 - 28). Because of the additive structure of each class’s covariance matrix, for every example’s gradient in the batch, we can independently sample one random Gaussian vector with rank-1 covariance and also another spherical Gaussian vector and add both vectors to this gradient for perturbation. This step would take $O(d)$ for each example and thus takes $O(Bd)$ in total.
>
> As a result, the entire Marvell algorithm has a time complexity of $O(Bd)$. This can be further sped up through parallel computation using multi-threading/multi-core. Considering that backpropagation through the cut layer would also require $O(Bd)$ time complexity, the Marvell algorithm would not slow down the split-learning process in any significant way.
>
> Empirically, in the table below, we present the average amount of time it takes to run Marvell for the three models we considered in our experiment section with the privacy hyperparameter value s = 4 (privacy protection as shown with purple line in Figure 3, 5, 6). We also compare it to the average time it takes to run one update iteration. (We additionally include the 95% confidence interval for the mean estimator of the run time.) Here we notice that the total run time of Marvell only takes up a very small amount of the total training time for each method, further corroborating our algorithm’s time efficiency advantages.
>
> |    | Batch size $B$ | cut layer feature dimension $d$ | average $\texttt{Marvell}$ run time (seconds/iteration) | |average update time (seconds/iteration)|
> |---| --- | --- | --- | --- | --- |
> | ISIC | $128$ | $1600 $ | $1.79 \times 10^{-2} \pm 7.03 \times 10^{-5}$ | | $3.94 \times 10^{-1} \pm 4.05 \times 10^{-3}$ |
> | Criteo | $1024$ | $128$ | $1.68 \times 10^{-2} \pm 8.67 \times 10^{-5}$ | | $4.84 \pm 3.28 \times 10^{-1}$ |
> | Avazu | $32768$ | $128$ | $1.75 \times 10^{-2} \pm 1.47 \times 10^{-4}$ | | $6.22 \pm 3.71 \times 10^{-1}$ |

---

### Official Review · Reviewer_ercK · 2021-11-01

**Correctness:** 3
**Technical Novelty And Significance:** 3
**Empirical Novelty And Significance:** 3
**Recommendation:** 8
**Confidence:** 4

**Main Review:**

Strengths:

* The paper studies an interesting problem.

* The paper is well written and organized.

Weaknesses:

* The setup for attack models (e.g., observations in attacks 1 and 2 in Section 3.3) seem very restrictive, and may only work for a specific set of binary classification problems with highly imbalanced classes, such as the disease prediction example given in the paper. What if the labels are more homogeneous, what would be the impact on observation 1?

* The privacy objective is defined as a function of KL divergence with respect to any underlying probability distribution on the perturbed gradients. This appears as a rather difficult objective to achieve in general. In fact, the paper then assumes a Gaussian distribution as the underlying distribution (and the perturbations) for each of the classes while evaluating the objective function. This assumption is not clear. The first thought that comes to mind is the central limit theorem, but then, as mentioned in the paper, one class may have very few samples (e.g., the disease prediction example). Moreover, it appears that the empirical mean of each class is treated as the true mean. However, the validity of this assumption would also depend on the number of samples collected for each class, which could be highly imbalanced between the two classes according to the motivated problem setup. This, combined with the fact that the max_norm heuristic seems to perform as well as Marvell, seems to require a stronger motivation for Marvell.







**Summary Of The Paper:**

This paper formulates a threat model on two-party split learning (parties have different features, with one party holding the labels) for binary classification, and provides insights about how simple functions on the gradients can be used to extract confidential label information. The authors then proceed with defenses to these attacks, based on random perturbations of the communicated gradients from one party to another. In doing so, the authors motivate a new measure for privacy termed the leak AUC, based on the AUC of the ROC curve.

**Summary Of The Review:**

Understanding the label leakage problem and protection against it under a split learning setup has the potential to start an intriguing line of work. The main drawback of this work is the restricted setup on which the attack/defense models are built on.

---

> ### Author Response · Authors · 2021-11-17
> **Response to Reviewer ercK Part 2/2**
>
> **[Use of Gaussian distribution]** We make two Gaussian distribution assumptions in our paper, which we explain in greater detail below.
>
> 1. We assume the unperturbed class-conditional gradient distribution to be Gaussian. Here this assumption is made to simplify our protection optimization problem: as we have derived a symmetrized KL divergence upper bound of our worst-case leak AUC objective, assuming spherical Gaussian distribution can directly gives us an analytical expression of the KL divergence which as we have shown reduces the optimization problem to a constant time optimization problem over 4 variables. Despite this simplifying assumption, we have empirically observed that the derived protection technique $\texttt{Marvell}$ can already provide strong protections against our identified attacks measured through our proposed leak AUC metrics. It is conceivable that this Gaussian distribution approximation could become not accurate enough to achieve strong protection performance against some new attack methods discovered in the future, but our worst case privacy protection formulation in Section 4.2 can still be used. Better optimizing this objective might require deriving tighter upper bounds for this worst-case leak AUC objective or finding ways to optimize the symmetrized KL divergence without an analytical expression like we have for Gaussian, which are interesting future directions motivated by our current work.
>
> 2. We additionally constrain the class-conditional noise perturbation distribution to be Gaussian. In principle, we can apply any random perturbation with arbitrary distributions to the gradients. A more flexible, larger search space of perturbation distributions could make privacy better while retaining the same utility. However, a larger perturbation distribution search space is difficult both to parameterize and to optimize over, potentially making the per-iteration optimization problem take much longer to solve. This could in turn slow down the split training considerably. In our problem, we have chosen a narrow class of perturbation distributions (gaussian distributions) to enable efficient parameterization and optimization. As we have seen in our experiments, only searching over Gaussian perturbations can already provide us with strong protection guarantees against our identified attacks. Finally, as explained in Appendix A.3 Remark (**Interpreting the optimal $Σ^∗_1$ and $Σ^∗_0$**), having a Guassian distribution assumption allows us to understand what type of noise structure is needed to obfuscate the gradient distributions. The solution’s two-component structure also enables us to efficiently sample the actual perturbation to be applied to each individual gradient.
>
>
> **[Empirical mean of each class as the true mean]** Thanks for bringing this up; in this particular instance, the empirical mean is in fact equivalent to the true mean. In particular, at any communication round, only the gradients of the examples contained in the batch are revealed from the label-party to the non-label party. In our threat model, only the labels of the examples in this batch can be leaked, and our protection objective is to ensure that a worst-case scoring function cannot be used to distinguish the positive and negative examples contained in this given batch. As we are only considering this one batch, the positive and negative gradient distribution $P^{(1)}$ and $P^{(0)}$ can be understood as the empirical distribution of the positive and negative gradients in the batch. The means can thus be directly computed (instead of estimated) by using the actual gradients in the batch.
>
> **[$\texttt{max}$_$\texttt{norm}$ vs $\texttt{Marvell}$]** Despite $\texttt{max}$\_$\texttt{norm}$ being a competitive baseline, one downside is that it does not take utility into consideration in its formulation and as a result cannot provide a flexible tradeoff between privacy and utility like $\texttt{Marvell}$ (as shown in Figure 4). In addition, it is motivated only through our identified attack methods; thus its protection may be much less effective against other, currently unknown practical attack methods. In contrast, $\texttt{Marvell}$ is formulated through a principled min-max privacy objective and as a result should be able to protect against attack methods beyond our two identified attacks. More generally, although $\texttt{max}$_$\texttt{norm}$ may work well in some practical scenarios, we believe there is value in deriving a principled protection technique like $\texttt{Marvell}$ instead of merely relying on a heuristic.

---

> > ### Comment · Reviewer_ercK · 2021-11-26
> > **Follow-up**
> >
> > Thank you, the responses addressed most of my comments so I increased my score. Regarding the Gaussian assumption on the gradients, I am still curious to see if there is a stronger motivation other than making the KL divergence computation simpler (maybe by empirically observing the distribution).

---

> ### Author Response · Authors · 2021-11-17
> **Response to Reviewer ercK Part 1/2**
>
> We thank the reviewer for their time in reviewing our submission. We respond to the points raised below:
>
> **[Restrictiveness of attack model]** When formulating our attack model, we describe four observations that could allow the non-label party to steal the private label with a high success rate. We describe the practical likelihood of these scenarios below. It is worth noting that these scenarios can occur even with perfectly balanced data (we provided an example of this in our original submission in Appendix A.2). We have highlighted this in our revision.
>
> - Observation 1 uses the fact that the model learned during training has less confidence about a positive example’s prediction than a negative example’s prediction. Although this can commonly happen when the data is imbalanced, it may **also happen with perfectly balanced data**. We provide an example of this in Appendix A.2. Discrepancies in model confidence between classes are prevalent in practice (including but not limited to the three benchmarks [ISIC, Criteo, Avazu] presented in our paper).
>
> - Observation 2 simply states that the 2-norm of the gradient vector $||\nabla_z h(z)\vert_{z = f(X)}||_2$ is on the same order of magnitude for both the positive and the negative class. This observation is very general and is applicable to a wide range of binary classification problems beyond those considered in our paper.
>
> - Observation 3 considers the different signs of $(\widetilde{p}_1 - y)$ for positive and negative examples and is **always** true for any model learned on any binary classification problem.
>
> - Observation 4 considers the positivity of the cosine similarity $\cos(\nabla_z h(z)\vert_{z = f(X_a)}, \nabla_z h(z)\vert_{z = f(X_b)})$. As we have mentioned in our discussion of Observation 4 in the paper, for neural networks with monotonically increasing activation functions (which would constitute the majority of the neural network use cases), this phenomenon would hold true.
>
> Beyond these observations, for the direction-based attack with cosine similarity, we have discussed how the adversarial non-label party can infer the labels using a majority counting attack. Even here we do not require the problem to have highly-imbalanced classes -- the attack would work as long as there is a reasonably high probability that the majority class has more examples than the minority class in a randomly sampled update batch. For example, when classes are only minorly imbalanced (45% positive and 55% negative), a batch of 128 iid sampled examples would have more positive examples (>64) than negative examples with less than 12% probability. Thus in the remaining 88% of sampled batches, our proposed majority counting direction-based attack would still succeed in fully recovering the label of every example in the batch.

---

### Author Response · Authors · 2021-11-17
**Common Response to All Reviewers**

We thank all reviewers for their time and thoughtful reviews. We have updated our main paper and appendix. For ease of comparison, we color any changed text and the newly added appendix section’s headers in purple. Below we first respond to a shared concern related to imbalanced classes and then address specific comments from each reviewer in separate responses.

**[Class imbalance]** We wish to clarify that our work is not limited to class-imbalanced data. In particular:

1. In terms of label leakage (attacks), as we have already explained in Appendix A.2, Observation 1 can occur even in perfectly class-balanced learning scenarios. Thus the norm-based attack can still work even in class-balanced learning scenarios whenever Observation 1 holds true.

2. In terms of label protection, our protection method Marvell is general and does not require the classification problem to be class-imbalanced to work.

We have highlighted these points in our revision.

---

### Decision · Program_Chairs · 2022-01-20

**Decision:**

Accept (Poster)

**Comment:**

All reviewers concur on the fact that the paper contains solid ideas. The discussion helped clarify the case of class-imbalance and no major concerns remained after discussion phase. I thank the authors for the additional details on execution time / complexity.

On a separate note and perhaps to dig further in the paper's ideas,

1- the validity of the Gaussian assumption carried in the paper was raised (e.g. ercK), but I would like to point out that Theorem 2 can also be derived for general exponential families given the objective in (2), with perhaps a reformulation of the trace constraint (still, this would imply the knowledge of the exponential family for the KL divergence to simplify).

2- when it comes to protecting labels, the authors might want to have a look at the rich literature on learning from label proportions, which shows that the knowledge of the class is not necessary to learn a supervised model (see for example Patrini et al, NeurIPS / NIPS 2014). Thus, protecting the class could in fact be more achievable than by just considering that learning “needs observed classes”.